# Homo-composition and hetero-structure nanocomposite *Pnma* Bi$_2$SeS$_2$ - *Pnnm* Bi$_2$SeS$_2$ with high thermoelectric performance

Bushra Jabar[1,5], Fu Li [1,5✉], Zhuanghao Zheng[1,5], Adil Mansoor[2], Yongbin Zhu[3], Chongbin Liang[1], Dongwei Ao[1], Yuexing Chen[1], Guangxing Liang[1], Ping Fan [1✉] & Weishu Liu [3,4✉]

Nanocomposite engineering decouples the transport of phonons and electrons. This usually involves the in-situ formation or ex-situ addition of nanoparticles to a material matrix with hetero-composition and hetero-structure (*he*C-*he*S) interfaces or hetero-composition and homo-structure (*he*C-*ho*S) interfaces. Herein, a quasi homo-composition and hetero-structure (*ho*C-*he*S) nanocomposite consisting of *Pnma* Bi$_2$SeS$_2$ - *Pnnm* Bi$_2$SeS$_2$ is obtained through a Br dopant-induced phase transition, providing a coherent interface between the *Pnma* matrix and *Pnnm* second phase due to the slight structural difference between the two phases. This *ho*C-*he*S nanocomposite demonstrates a significant reduction in lattice thermal conductivity (~0.40 W m$^{-1}$ K$^{-1}$) and an enhanced power factor (7.39 μW cm$^{-1}$ K$^{-2}$). Consequently, a record high figure-of-merit $ZT_{max} = 1.12$ (at 773 K) and a high average figure-of-merit $ZT_{ave} = 0.72$ (in the range of 323–773 K) are achieved. This work provides a general strategy for synergistically tuning electrical and thermal transport properties by designing *ho*C-*he*S nanocomposites through a dopant-induced phase transition.

[1] Shenzhen Key Laboratory of Advanced Thin Films and Applications, College of Physics and Optoelectronic Engineering, Shenzhen University, 518060 Shenzhen, China. [2] Faculty of Materials and Manufacturing, Beijing University of Technology, 100 Peenle Yuan, Chaoyang District, 100124 Beijing, China. [3] Department of Materials Science and Engineering, Southern University of Science and Technology, 518055 Shenzhen, China. [4] Guangdong Provincial Key Laboratory of Functional Oxide Materials and Devices, Southern University of Science and Technology, 518055 Shenzhen, China. [5] These authors contributed equally: Bushra Jabar, Fu Li, Zhuanghao Zheng. ✉email: lifu@szu.edu.cn; fanping@szu.edu.cn; liuws@sustech.edu.cn

Thermoelectric (TE) technology, which directly converts heat into electricity, is a potential solution for securing an affordable green energy source by harvesting large-scale mid-grade waste heat (at/near mid-range temperatures)[1,2]. The dimensionless figure-of-merit ($ZT = \sigma S^2 T/\kappa$) and power factor ($PF = \sigma S^2$) are the material-level performance scales for energy conversion efficiency and output power factor, where $S$, $\sigma$, $\kappa$, and $T$ are the Seebeck coefficient, electrical conductivity, thermal conductivity, and temperature, respectively[3,4]. Generally, $ZT$ can be improved by lowering $\kappa$ or enhancing $PF$. However, simultaneously improving $PF$ while reducing the value of $\kappa$ is a significant challenge. Nanocomposite engineering, a vital strategy for decoupling the transport of phonons and electrons, involves the in-situ formation and ex-situ addition of nanoparticles to a material[5–14]. The composition difference between the matrix and second phases can induce mass fluctuation, strains[15–17], or dislocation[18,19] at their interfaces, which can dramatically scatter phonons and reduce lattice thermal conductivity. A nanocomposite with coherent nanoinclusions (e.g., hetero-composition and homo-structure (heC-hoS)[13,18,20,21], PbTe-AgSbTe_2[22]) suffers less charge carrier mobility reduction than one with incoherent nanoinclusions (e.g., hetero-composition and hetero-structure (heC-heS)[7,10,23], PbTe-Ag_2Te[24]). Zhao et al. suggested that the band alignment between the matrix and second phases could play a critical role in minimizing electron scattering[25].

Furthermore, extremely low thermal conductivities have been observed in various materials near their phase transition, caused by both pressure-induced and temperature-induced transitions. Near the phase transition point, a material can be considered as a metastable nanocomposite with a homo-composition and a hetero-structure. So far, some research has been carried out to investigate the structural changes with the symmetry reduction of specific phase change materials and their effect on TE properties under the influence of pressure or temperature[26–31]. However, the structural phase transitions and underlying driving force (temperature or pressure) of such materials have not yet been systematically investigated due to their undefined intermediate structures and internal atomic distortions. It is also very difficult to obtain precise regulation in pressure-induced transition, while temperature-induced transition may only be effective for the structural transition of specific materials in a narrow temperature range[26,31–34].

Herein, we propose a quasi homo-composition and hetero-structure (hoC-heS) nanocomposite composed of Pnma Bi_2SeS_2 - Pnnm Bi_2SeS_2, which provides a coherent interface between the matrix and second phase due to the slight structural difference between the two phases. Br element is familiarly donor in the Bi_2S_2, Bi_2Se_3, and Bi_2SeS_2[11,23]. However, it was found that the Br dopant caused the phase transition from Pnma Bi_2SeS_2 to Pnnm Bi_2SeS_2 in our work. And the initial orthorhombic Pnma phase also exhibits structural distortion. Due to the significant reduction in lattice thermal conductivity ($\kappa_L$) and low impact to carrier mobility ($\mu$) caused by the Br dopant, a record high figure-of-merit $ZT_{max} = 1.12$ (at 773 K) and a record average $ZT_{ave} = 0.72$ (in the range of 323–773 K) were achieved in the Bi_2SeS_2 family. It should be noted that the $ZT$ value of most reported sulfide compounds is lower than 1.0 due to the low carrier mobility and high lattice thermal conductivity so far, although sulfide compounds with cheaper, low-toxicity, and high earth abundance elements have been extensively studied in recent years. This work provides a general strategy for designing hoC-heS nanocomposites through a dopant-induced phase transition to enhance TE properties.

## Results

Figure 1 shows the structure of the as-fabricated Pnma Bi_2SeS_2 - Pnnm Bi_2SeS_2 quasi hoC-heS nanocomposite induced by the Br dopant. This structure is very different from those of previously

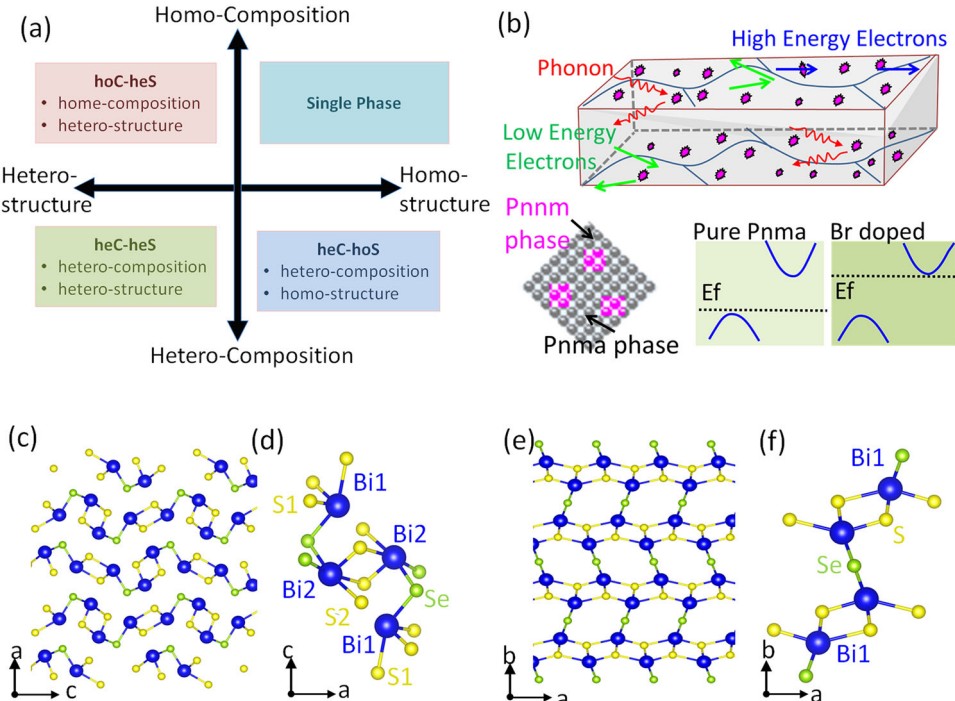

**Fig. 1 Schematic representation of the nanocomposition and crystal structure of Bi_2SeS_2. a** Summary of previously reported nanocomposites classified by the composition and crystal structure of their nanoinclusion and matrix. Most reported nanocomposites are hetero-composition and homo-structure (heC-hoS) or hetero-composition and hetero-structure (heC-heS). **b** Structure and schematic illustration of transport mechanism inside Pnma Bi_2SeS_2 - Pnnm Bi_2SeS_2 hoC-heS nanocomposite. **c–d** Crystal structure of Bi_2SeS_2 Pnma phase in the ac plane. **e–f** Crystal structure of Bi_2SeS_2 Pnnm phase in the ab plane.

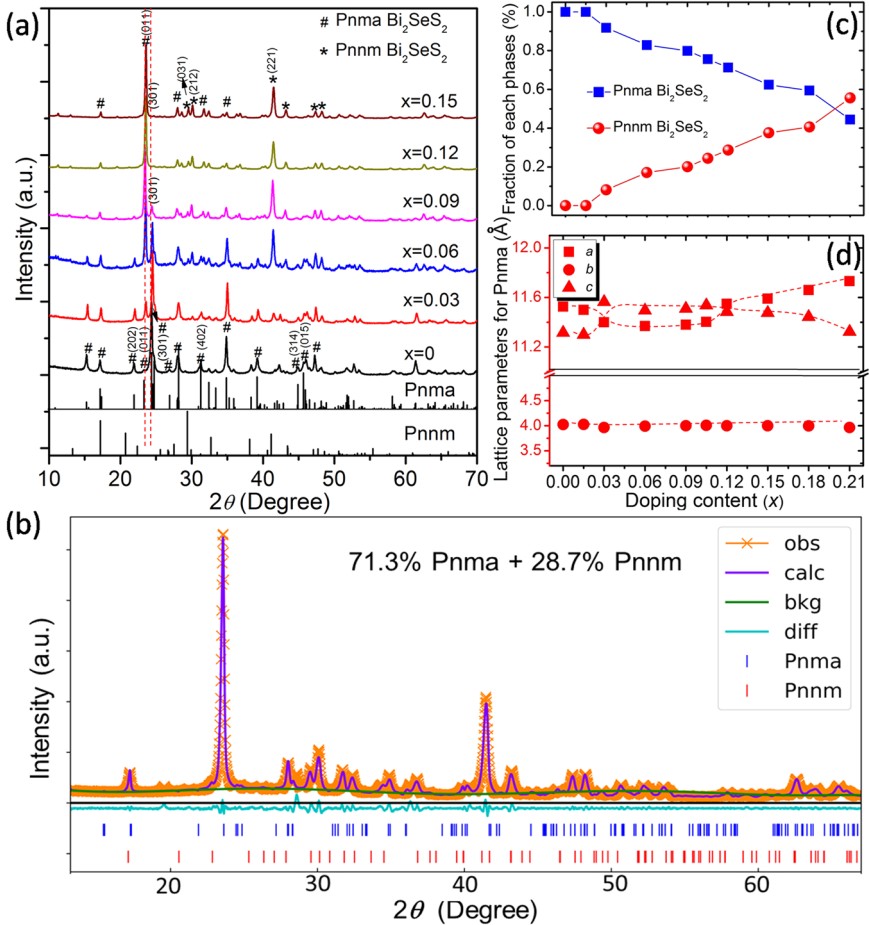

**Fig. 2 Phase structure and lattice parameters. a** XRD patterns of Bi$_2$Se$_{1-x}$Br$_x$S$_2$ ($x = 0$, 0.03, 0.06, 0.09, 0.12, 0.15). **b** Rietveld refinement analysis from XRD patterns of Bi$_2$Se$_{1-x}$Br$_x$S$_2$ ($x = 0.12$), where obs, calc, bkg, and diff represent the observed, calculated, background, and residual differences between the curves, respectively. **c** Pnma and Pnnm phase fractions in the sample as a function of Br content. **d** Lattice constant parameters of the Pnma phase as a function of Br content. The dashed lines are provided as a guide.

reported nanocomposites with a hetero-composition and homo-structure (heC-hoS) or hetero-composition and hetero-structure (heC-heS) (Fig. 1(a, b)). The matrix phase Pnma Bi$_2$SeS$_2$ demonstrates an orthorhombic layered structure with each quintuple layer assembled by weak Van der Waals interactions, a widely observed characteristic of polycrystalline Bi$_2$SeS$_2$. Five distinct atomic sites (Bi1, Bi2, S1, S2, and Se) are presented in the unit cell due to lattice symmetry operation (S-Bi-S(Se)-Bi-S), in which all the Bi and S (or Se) atoms occupy the 4c Wyckoff positions in the Pnma space group (Fig. 1(c))[9,35–37]. This layered assembly possesses a highly confined S-coordination environment around both the Bi1 and Bi2 cations with three short and four long bonds (Fig. 1(d)). However, a new Bi$_2$SeS$_2$ phase crystallized in a Pnnm orthorhombic structure can be found (Fig. 1(e, f)).

Figure 2 shows the XRD patterns of the hoC-heS nanocomposite with different Bi$_2$Se$_{1-x}$Br$_x$S$_2$ compositions ($x = 0$, 0.03, 0.06, 0.09, 0.12, and 0.15). The two phases (Pnma and Pnnm) appear when the Br content is higher than $x = 0.01$ (Fig. 2(a), Figs. S1 and S2: Bi$_2$Se$_{1-x}$Br$_x$S$_2$, $x = 0$–0.21). Based on the Reitveld refinement (Figs. 2(b) and S3), the XRD peaks match well with those of orthorhombic Pnma (space group: 62) and Pnnm (space group: 58). It should be noted that the Pnma phase is widely reported for polycrystalline Bi$_2$SeS$_2$ and Bi$_2$S$_3$ compounds, while the Pnnm phase is still a theoretically predicted structure[38,39]. A noticeable and continuous variation in the Pnma crystal structure can be observed with increasing Br content in the normal Bi$_2$Se$_{1-x}$Br$_x$S$_2$ composition. Figure 2(c) shows the molar fraction

evaluation of Pnma Bi$_2$SeS$_2$ and Pnnm Bi$_2$SeS$_2$. The fraction of Pnnm Bi$_2$SeS$_2$ increases with increasing Br content while Pnma Bi$_2$SeS$_2$ shows the opposite trend. For instance, Bi$_2$Se$_{1-x}$Br$_x$S$_2$ ($x = 0.12$) consists of 71.3% Pnma phase and 28.7% Pnnm phase, while Bi$_2$Se$_{1-x}$Br$_x$S$_2$ ($x = 0.21$) consists of 44.4% Pnma phase and 55.6% Pnnm phase.

In addition to the Br dopant-derived phase change, the relative intensity of the (301), (011), (202), (402), (314), and (015) peaks of the Pnma phase significantly change with increasing Br content, indicating continuous lattice distortion before the transition into the Pnnm phase. In addition, it is reported that their surface energies can quantitatively describe the stability of various surfaces[40]. The calculated formation energy along the plane (011) and (301) indicates that the formation energy along the plane (011) has reduced, while it has increased along the plane (301) after Br doping. This means the plane (011) might be easy to form during the preparation due to the low energy after Br doping. Therefore, it is clear that the Bi$_2$SeS$_2$ system experiences a two-fold structural evolution with increasing Br content. First, the initial orthorhombic Pnma phase exhibits some structural distortion, revealing planar transitions that preserve the unit cell and achieve complete orientation along the (011) plane when $0.09 \leq x \leq 0.12$. Second, doping-induced structural variations simultaneously promote inter-orthorhombic phase transformation from Pnma to Pnnm. Figure 2(d) shows the Br dopant-induced lattice parameter evolution. In the current Pnma phase, the b-axis does not show significant change. However, the a- and

c- axes approach each other and overlap without a significant expansion in the unit cell volume (Fig. S4 and Table S1) with increasing Br content from $x = 0$ to $x = 0.015-0.03$, indicating the absence of prominent structural disorder in the $Pnma$ phase. When x is further increased from 0.03 to 0.105, the unit cell parameters show similar constant behavior while the unit cell volume significantly expands (Fig. S4). As the Br content increases from $x = 0.12$ to $x = 0.18$, the $a$- and $c$- axes move away from each other with a continuous expansion in the unit cell volume (Fig. S4). These discontinuities or deviations in the lattice parameters of the Br-doped $Pnma$ phase $Bi_2SeS_2$ indicate the existence of lattice distortion. For the $Pnnm$ phase, the lattice parameters of $b$ and $c$ gradually reduce when the Br content x is higher than 0.105 (Fig. S5). However, the decrease is not significant for the samples with low Br content x ($x < 0.105$), mainly due to the lower dopant content and the slight difference in the ionic radius between $Br^-$ (1.96 Å) and $Se^{2-}$ (1.98 Å).

Figure 3(a–d) shows high-resolution TEM images of the as-fabricated $hoC$-$heS$ nanocomposite $Bi_2Se_{1-x}Br_xS_2$ ($x = 0.12$). The existence of the $Pnma$ and $Pnnm$ phases can clearly be seen. The $Pnnm$ phase has a typical irregular oval-like morphology and is coherently embedded inside the $Pnma$ phase grains. The inverse fast Fourier transform (IFFT) image shown in Fig. 3(b) displays a defect-free phase boundary between these two phases. In addition, energy dispersive X-ray spectroscopy (EDS) shows that the elements, including Br, are homogeneously distributed in the $Pnma$ and $Pnnm$ phases, with the exception of excess Bi observed in the $Pnnm$ phase (Fig. S6). Figure 3(e, f) shows high-angle annular dark field scanning transmission electron microscopy (HAADF-STEM) images of $Bi_2Se_{1-x}Br_xS_2$ ($x = 0, 0.12$) along the $c$-axis. Bi-centric atomic distributions are prominent, with well-arranged atoms along the $c$-axis of the undoped $Bi_2SeS_2$ (Fig. 3(e)). However, the Br dopant clearly induces atomic disarrangement and lattice distortion with uniform and distinctive elongation or contraction (Fig. 3(f)), with the doped materials displaying local phase structural variations without disturbing the symmetry. By comparing the peak intensity profiles of line 1 (or line 2) to line 3 (or line 4), it can be predicted that the significant contrast in strain causes atomic disarrangement within the frame structure of the orthorhombic $Pnma$ phase due to Br doping. The lattice variations along the $c$-axis are related to modifications of the bond distances. The calculated bond lengths indicate that the short Bi1- S1(S) and Bi2-S2(S) bonds and the long Bi1-S1(L) and Bi2-S2(L) bonds do not significantly change after doping (Fig. S7). However, the increased Bi1-Se bond length and decreased Bi2-Se bond length (Fig. S7) suggest that Br elongates the orthorhombic layered structure (Fig. 3). This lattice evolution with disordered bond length can lead to lattice anharmonicity, which is an important phonon scattering mechanism in TE materials. Figure 4(a–d) shows that the $Pnnm$ phase particles are tensile strained, with the maximum tensile strain reaching 3%. The strains in the $\varepsilon_{xx}$ and $\varepsilon_{yy}$ directions show continuous contrast across the grain boundary (Fig. 4(a–d)). The rotational strain profile ($\varepsilon_{rot}$) also shows continuous contrast across the phase boundaries (Fig. S8). This means that the strain at the interface is insignificant, suggesting that the corresponding phase boundary between the $Pnma$ and $Pnnm$ phases is defect-free.

Figure 5 show the electrical transport properties of as-fabricated $hoC$-$heS$ nanocomposite $Bi_2Se_{1-x}Br_xS_2$ ($x = 0, 0.015, 0.03, 0.06, 0.09, 0.105, 0.12, 0.15, 0.18,$ and 0.21). The transport properties change with increasing doping content x, and this change can be roughly divided into three parts. The undoped $Bi_2SeS_2$ exhibits poor electrical conductivity ($\sigma$) at room temperature due to its low carrier concentration ($n$) (Fig. 5(a)). However, the room temperature $\sigma$ value significantly increases from $0.25 \times 10^4$ $Sm^{-1}$ to $6.26 \times 10^4$ $Sm^{-1}$ with a low amount of

Br ($x = 0.015$), corresponding to an enhanced $n$ value from $1.28 \times 10^{19}$ to $46.6 \times 10^{19}$ $cm^{-3}$ (Fig. 5(b)). Thus, Br is a very effective dopant. However, the room temperature $\sigma$ decreases from $5.62 \times 10^4$ $Sm^{-1}$ to $2.42 \times 10^4$, $1.49 \times 10^4$, and then $0.59 \times 10^4$ $Sm^{-1}$ as the Br content further increases from $x = 0.03$ to $x = 0.06, 0.09,$ and 0.105 (Fig. 5(e)). For these doped nanocomposite samples, the $Pnnm$ phase appears and increases with increasing Br content but remains lower than 40%. A further increase in Br content (from $x = 0.105$ to 0.12 and 0.15) results in a slight increase in $\sigma$ (with slightly enhanced $n$ and moderate $\mu$), but $\sigma$ declines again when $x \geq 0.18$ (Fig. 5(h)). It means that when the $Pnnm$ phase fraction is higher than 40% and $x \geq 0.18$, the value of $\sigma$ deteriorates due to the decrease of $\mu$ (Fig. 5(a)). The nano $Pnnm$ phase would be a new electron scattering center. The temperature-dependent $\sigma$ exhibits two trends. When $0.015 \leq x \leq 0.03$, the value of $\sigma$ decreases monotonically with increasing temperature, acting as a degenerate semiconductor. In contrast, when $0.06 \leq x \leq 0.21$, the value of $\sigma$ first decreases and increases with increasing temperature, acting as a thermally activated semiconductor.

Figure 5(c), (f), and (i) shows the temperature-dependent Seebeck coefficient ($S$) of the as-fabricated $hoC$-$heS$ nanocomposite $Bi_2Se_{1-x}Br_xS_2$ ($x = 0, 0.015, 0.03, 0.06, 0.09, 0.105, 0.12, 0.15, 0.18,$ and 0.21), which presents good consistency with $\sigma$. The Br-free $Bi_2SeS_2$ nanocomposite shows a high room temperature $S$ value of $-300$ $\mu V$ $K^{-1}$. The $S$ values decrease at low Br doping concentrations ($0.015 \leq x \leq 0.09$) due to the higher $n$ values of these nanocomposites compared with the undoped nanocomposite. At high Br doping concentrations ($0.105 \leq x \leq 0.21$), the magnitude of $S$ increases to 200 $\mu V$ $K^{-1}$, and the bipolar transport shifts to lower temperatures. By using the $S$ and $n$ data and assuming that acoustic phonon scattering is prominent in the $Bi_2SeS_2$ matrix, a Pisarenko relation (at 300 K) can be obtained for $Bi_2Se_{1-x}Br_xS_2$ (Fig. S9). The $S$ values of the Br-doped samples deviate from the Pisarenko relation when $x > 0.06$, suggesting that the electronic density of states N(E) changes due to activation of the conduction bands (CBs) via doping. In addition, the density of state (DOS) effective mass ($m^*_d$) was obtained using the equations (S2)–(S4) in the Supporting Information, assuming the as-fabricated nanocomposite as quasi-single uniform phase (Fig. S9). A increased $m^*_d$ from $1.62m_e$ ($x = 0$) to $6.90m_e$ (at $x = 0.21$) with Br content of $Bi_2Se_{1-x}Br_xS_2$ was observed. The variations in $m^*_d$ values with Br content results from the band structure change and the interface effect. The enhanced $\sigma$ and moderate $S$ exhibited by all the doped samples result in a remarkable enhancement in their power factors (PF) across the whole temperature range (Fig. 5(d), (g), and (j)). The largest PF of $\sim 7.39$ $\mu W$ $cm^{-1}$ $K^{-2}$ at 773 K is achieved by $Bi_2Se_{1-x}Br_xS_2$ ($x = 0.12$). This value is seven times higher than that of pure $Bi_2SeS_2$ and is also the highest value reported thus far among $Bi_2SeS_2$ and $Bi_2S_3$-based materials reported in the literature[8–11].

Figure 6 illustrates the evolution of the Br-free and Br-doped $Bi_2SeS_2$ band structure in both the $Pnma$ and $Pnnm$ phases. First, Fig. 6(a–d) shows the Br-free and Br-doped $Bi_2SeS_2$ $Pnma$ phase band structures, both of which are direct bandgap semiconductors with conduction band minima (CBM) and valence band maxima (VBM) near the G point along the high symmetric line of the Brillouin zone. However, the CBM and VBM of the Br-doped $Bi_2SeS_2$ are not as sharp as those of the pure structure, and significant band convergence can be observed due to doping. Moreover, the Fermi level is prominently inclined toward the CB in Br-doped $Pnma$ $Bi_2SeS_2$, confirming the Br is an effective donor. According to the relationship between the electron and hole concentrations and bandgap[41], Br dopant increases the electron concentration but decreases the hole concentration. In short, in the Br-doped $Bi_2SeS_2$ structure, the CB calculations

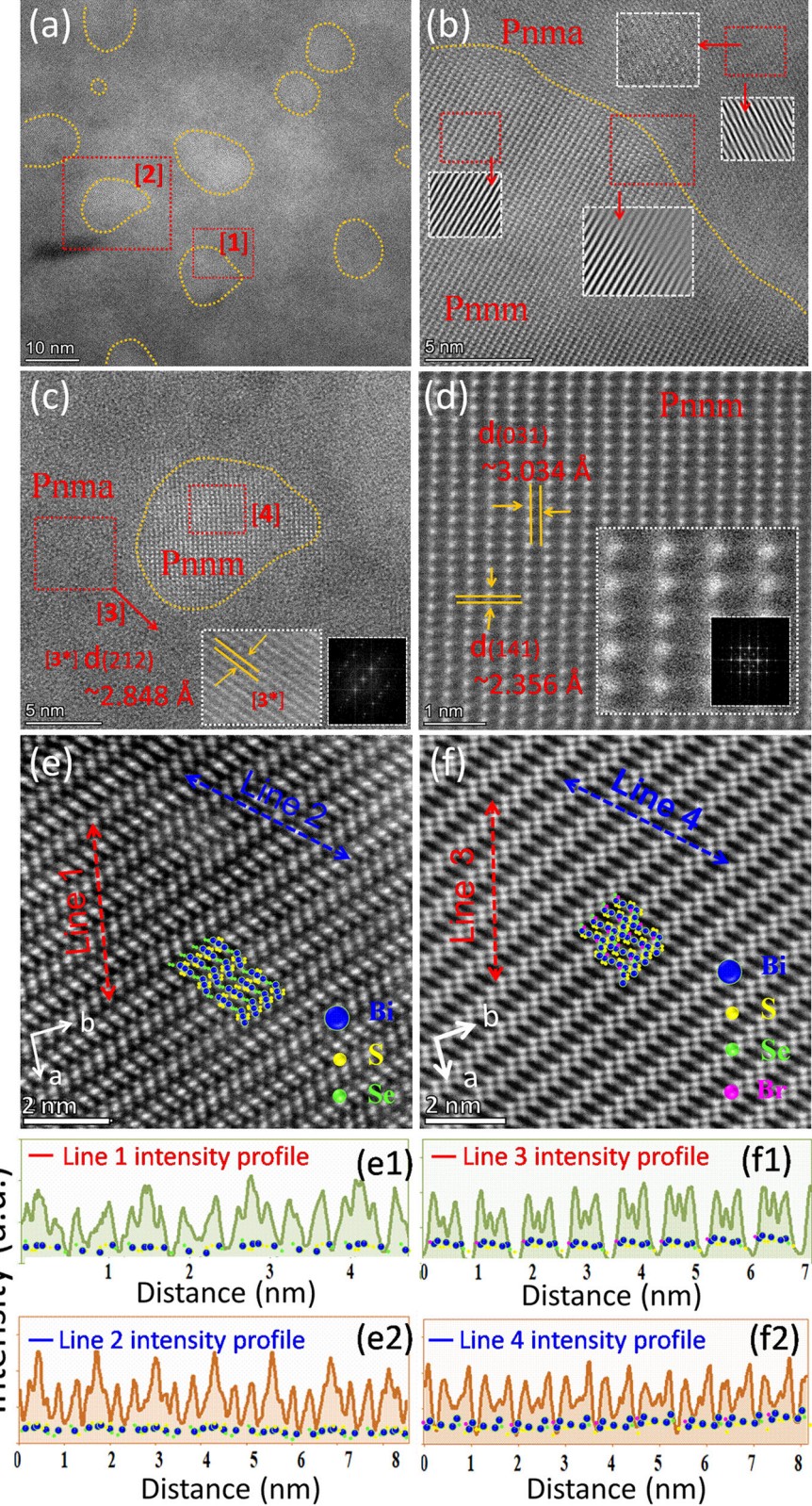

**Fig. 3 Structural characterization. a** TEM micrograph of Bi$_2$Se$_{1-x}$Br$_x$S$_2$ ($x = 0.12$), where the yellow dotted regions show the *Pnnm* nanophase. **b** HRTEM micrograph of the red dotted square [1] in (**a**). **c** HRTEM micrograph of the red dotted square [2] in (**a**), where the inset is the HRTEM micrograph of region [3]. **d** HRTEM micrograph of the red dotted square [4] in (**c**). **e–f** STEM-HAADF images of undoped and Br-doped Bi$_2$SeS$_2$ samples showing the orthorhombic *Pnma* structure along the *c*-direction. (e1)-(e2) Intensity line profiles corresponding to line 1 and line 2 in (**e**). (f1)-(f2) Intensity line profiles corresponding to line 3 and line 4 in (**f**).

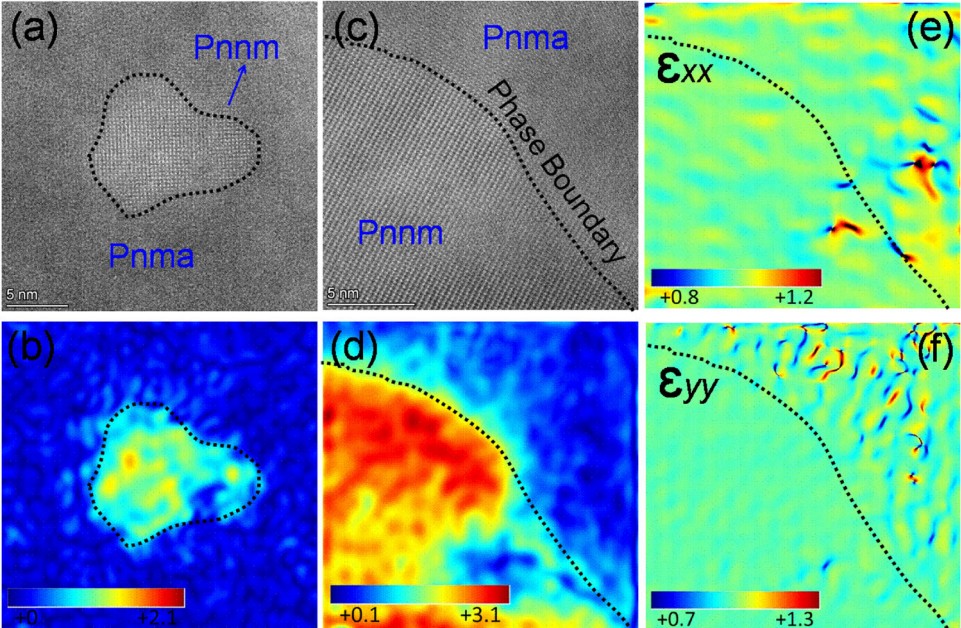

**Fig. 4 Geometric phase analysis (GPA) of corresponding HRTEM images from Fig. 3 of Bi₂Se₁₋ₓBrₓS₂ (x = 0.12).** GPA strain maps (**b**) and (**d**) are related to (**a**) and (**c**) and the color bars show the induced strain in the *Pnnm* phase. **e–f** Corresponding strain maps along the $\varepsilon_{xx}$ and $\varepsilon_{yy}$ directions for (**c**).

(with impurity levels in the bandgap) indicate that more free electron carriers are generated once Br is doped into the $Bi_2SeS_2$ lattice, with Br acting as a donor. The introduced overcharge effect in this Br-doped system, due to defect states, indicates that modifications can be made in the structure to utilize the overcharge. In addition, the partial density of states (DOS) of Bi primarily contributes to the total DOS in the CBM for both the *Pnma* and *Pnnm* phases (Fig. 6(c) and (g)). The DOS in the CBM along the *y*-axis shows that the intensity of peaks in the *Pnnm* phase $Bi_2SeS_2$ is reduced, which can be mainly ascribed to lattice distortions (Fig. 6(f)). This results in a reduction in energy. Here, distortions yield an orthorhombic *Pnnm* $Bi_2SeS_2$ structure with a slightly reduced symmetry. Overall, these results indicate that the Br dopant acts as a donor, causing significant structural distortion and the inter-orthorhombic transformation of *Pnma* to *Pnnm*. This phase transformation optimizes the electrical transport properties of $Bi_2SeS_2$. Besides, the calculated formation energy also shows that the formation energy value for Br doping at Se site is much lower than that at S site (Fig. S10(a)). This means Br should prefer to occupy the Se site in $Bi_2SeS_2$.

Figure 7 shows the thermal conductivity and *ZT* values of as-fabricated *hoC-heS* nanocomposite of *Pnma* $Bi_2SeS_2$ - *Pnnm* $Bi_2SeS_2$ in a nominal composition of $Bi_2Se_{1-x}Br_xS_2$ (x = 0, 0.09, 0.12, 0.15, 0.21). Other $Bi_2Se_{1-x}Br_xS_2$ with x = 0.015, 0.03, 0.06, 0.105, and 0.18 are shown in the Fig. S11. The undoped $Bi_2SeS_2$ shows a lower $\kappa$ of 0.72 W m⁻¹ K⁻¹ (Fig. 7(a)). While the value of $\kappa$ increases with a very low amount of Br (x = 0.015) due to the contribution from carriers (Fig. S11(a)), it gradually decreases to 0.60 W m⁻¹ K⁻¹ with an increase in Br content to x = 0.09 and further reduces to ~0.5 W m⁻¹ K⁻¹ when x ≥ 0.12. Furthermore, the value of $\kappa$ shows temperature-independent behavior for the heavily doped samples (0.09 ≤ x ≤ 0.21), maintaining a minimum at ~725 K and then slightly shifting to higher values with a further increase in temperature due to the bipolar effect. As shown in Fig. 7(b), the value of $\kappa_L$, which can be estimated by subtracting the electronic thermal conductivity $\kappa_e$ from $\kappa$ ($\kappa_L = \kappa - \kappa_e$, where $\kappa_L$ is the lattice thermal conductivity and $\kappa_e$ is the electronic thermal conductivity (Fig. S11(b))), is ~0.72 W m⁻¹ K⁻¹ at room temperature for pure $Bi_2SeS_2$. The value of $\kappa_L$ declines with

increasing temperature until 550 K, then increases due to the bipolar effect. However, for a Br content of x = 0.09, the value of $\kappa_L$ at room temperature is only 0.52 W m⁻¹ K⁻¹. This further declines to ~0.40 W m⁻¹ K⁻¹ for x = 0.12. For the Br-doped sample where x ≥ 0.15, the value of $\kappa_L$ slightly increases to ~0.47 W m⁻¹ K⁻¹. The contribution of the bipolar effect becomes negligible with increasing Br content (Fig. S11(d)). Because of this, the value of $\kappa_L$ of the heavily doped samples is temperature-independent (Fig. 7(b)). The theoretical lattice thermal conductivity was calculated via the Debye-Callaway model by combining all the substantial factors (presented in the Supporting Information). In the $Bi_2Se_{1-x}Br_xS_2$ system, the dominant phonon-scattering mechanisms involve scattering processes from point defects/alloy elements, nanophase, phonon-phonon Umklapp scattering, boundaries between the *Pnma* and *Pnnm* phases, and electron-phonon interactions. Figure 7(b) shows that the calculated value of $\kappa_L$ is in good agreement with the experimental results before the onset of the bipolar effect, which is the main reason for the deviation between the calculated $\kappa_L$ and the experimental data. Parameter A (Table S3), which is the preset parameter for point defects (including nanophase boundaries) in the expression of total phonon relaxation time, shows a reduction due to doping. However, parameter A increases from 3.48 to 10.4 as x increases from 0.015 to 0.12, which indicates enhanced phonon blocking due to the presence of intensive nanophase boundaries or impurity centers. A further increase in x from 0.15 to 0.21 causes a decrease in the value of A (Table S3). This explains the increase in $\kappa_L$ when x ≥ 0.15. At high values of x, the nanophase boundaries between the *Pnma* and *Pnnm* phases agglomerate due to the high density of *Pnnm* nanoinclusions, which causes a reduction in the number of effective phonon scattering centers and increases the value of $\kappa_L$. Therefore, x = 0.12 is the optimum Br dopant level with the appropriate *Pnnm* phase fraction in this nanocomposite.

Due to the combination of a significantly enhanced *PF* and reduced $\kappa$, a record high $ZT_{max}$ of 1.12 (at 773 K) and a record high average $ZT_{ave}$ of 0.72 (in 323–773 K) are achieved for the doped $Bi_2Se_{1-x}Br_xS_2$ system when x = 0.12 (Fig. 7(c)). The average $ZT_{ave}$ is calculated by integrating the area under the *ZT* curves in

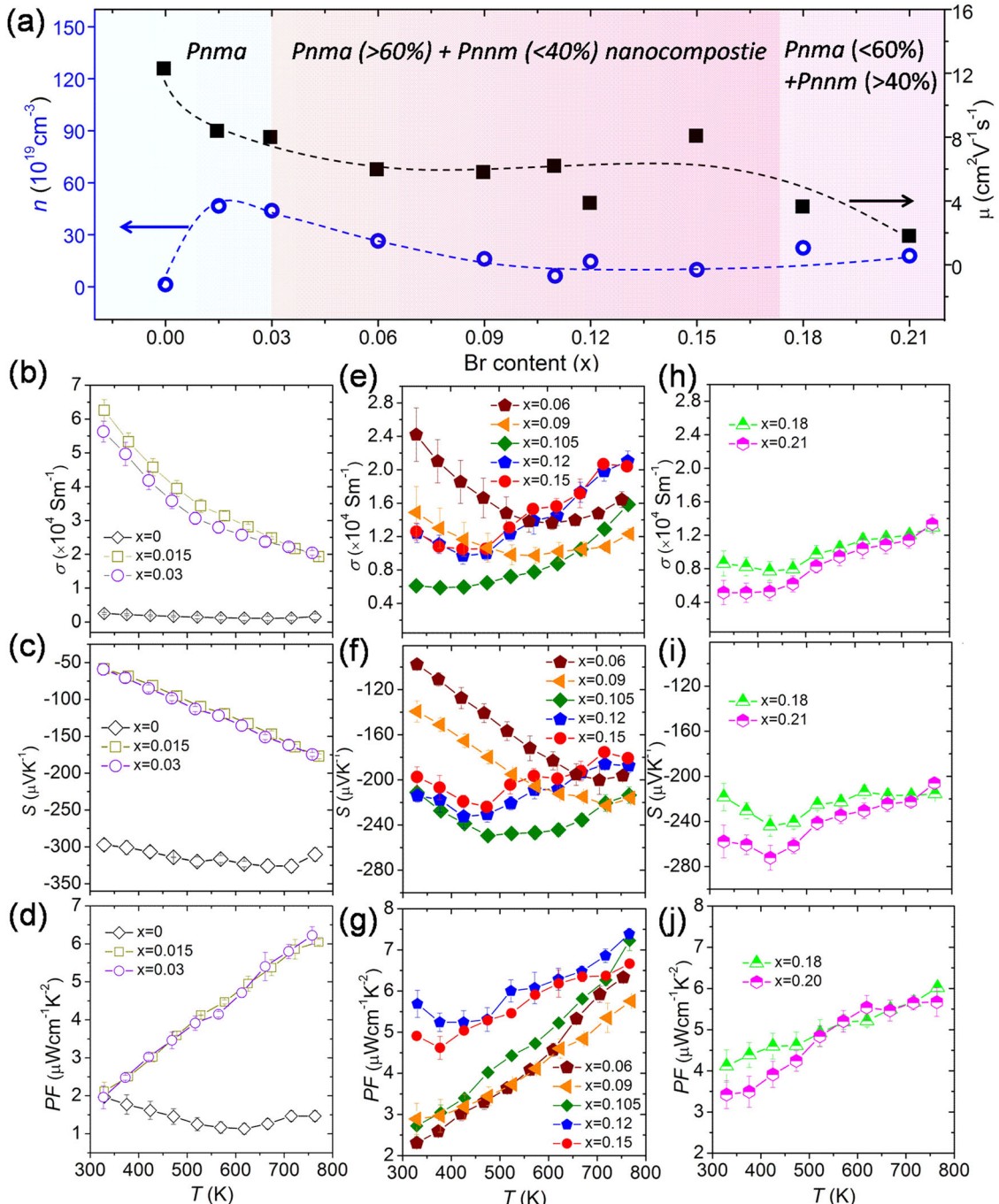

**Fig. 5 Electrical transport properies of the fabricated materials. a** Variation of carrier concentration ($n$) and mobility ($\mu$) (at 300 K) with Br content. The dashed lines are provided as a guide. **b–d** Temperature dependence of electrical resistivity ($\sigma$), Seebeck coefficient ($S$), and power factor (PF) of $Bi_2Se_{1-x}Br_xS_2$ ($x = 0$, 0.015, 0.03). **e–g** Temperature dependence of electrical resistivity ($\sigma$), Seebeck coefficient ($S$), and power factor (PF) of $Bi_2Se_{1-x}Br_xS_2$ ($x = 0.06$, 0.09, 0.105, 0.12, 0.15). **h–i** Temperature dependence of electrical resistivity ($\sigma$), Seebeck coefficient ($S$), and power factor (PF) of $Bi_2Se_{1-x}Br_xS_2$ ($x = 0.18$, 0.21). Error bars were estimated from the repeatability of the experimental result; three measurements were carried out for each material.

Fig. 7(c) according to the following formula,

$$ZT_{ave} = \frac{1}{T_h - T_c} \int_{T_c}^{T_h} ZT \, dT \qquad (1)$$

where $T_c = 323$ K and $T_h = 773$ K are the hot-side and cold-side temperatures. These TE transport properties show good reproducibility and thermal stability (Fig. S12). The optimized nanocomposite of $Bi_2Se_{1-x}Br_xS_2$ is composed of 71.3% *Pnma* $Bi_2SeS_2$ and 28.7% *Pnnm* $Bi_2SeS_2$. When the *Pnnm* phase fraction increases, the

value of $ZT$ decreases (Fig. 7(d)) due to deterioration of electrical conductivity and a slight enhancement of thermal conductivity, demonstrating that a moderate amount of *Pnnm* phase in the nanocomposite is ideal for achieving good TE transport properties. As a comparison, Table S4 and Fig. S13 present the TE properties, including $\sigma$, $S$, $\kappa$, and $ZT$ value, of some representative $Bi_2S_3$ related systems. Both the $ZT_{max}$ (1.12 at 773 K) and $ZT_{ave}$ (0.72 in 323–773 K) values of as-fabricated *ho*C-*he*S nanocomposite of *Pnma* $Bi_2SeS_2$ - *Pnnm* $Bi_2SeS_2$ are much higher than previously reported $Bi_2SeS_2$ and $Bi_2S_3$-based materials, and other reported sulfide

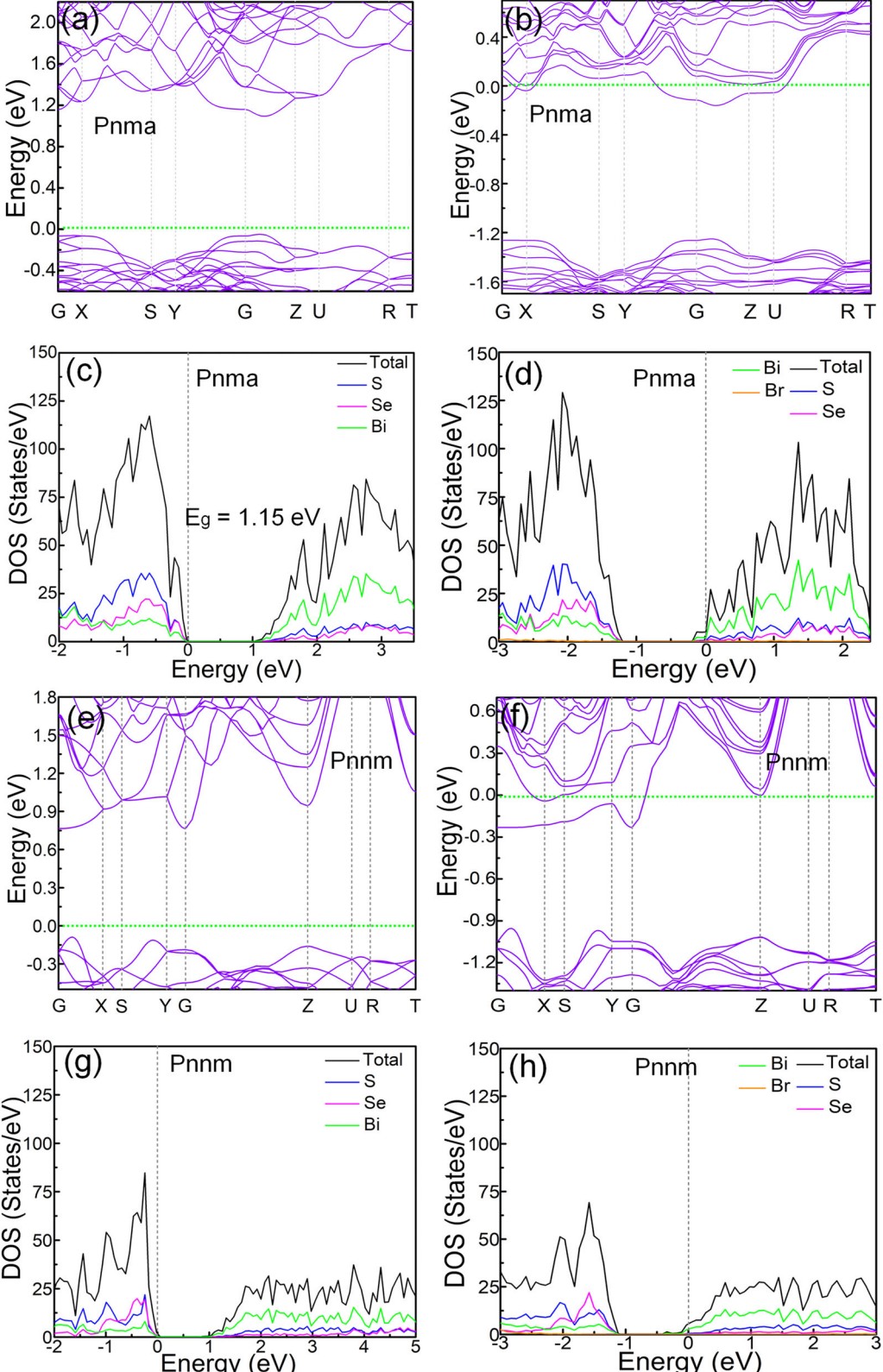

**Fig. 6 Electronic band structure and density of state (DOS) calculations. a** and **c** Electronic band structure and DOS of pure $Bi_2SeS_2$ with *Pnma* structure. **b** and **d** Electronic band structure and DOS of Br-doped $Bi_2SeS_2$ with *Pnma* structure. **e** and **g** Electronic band structure and DOS of pure $Bi_2SeS_2$ with *Pnnm* structure. **f** and **h** Electronic band structure and DOS of Br-doped $Bi_2SeS_2$ with *Pnnm* structure.

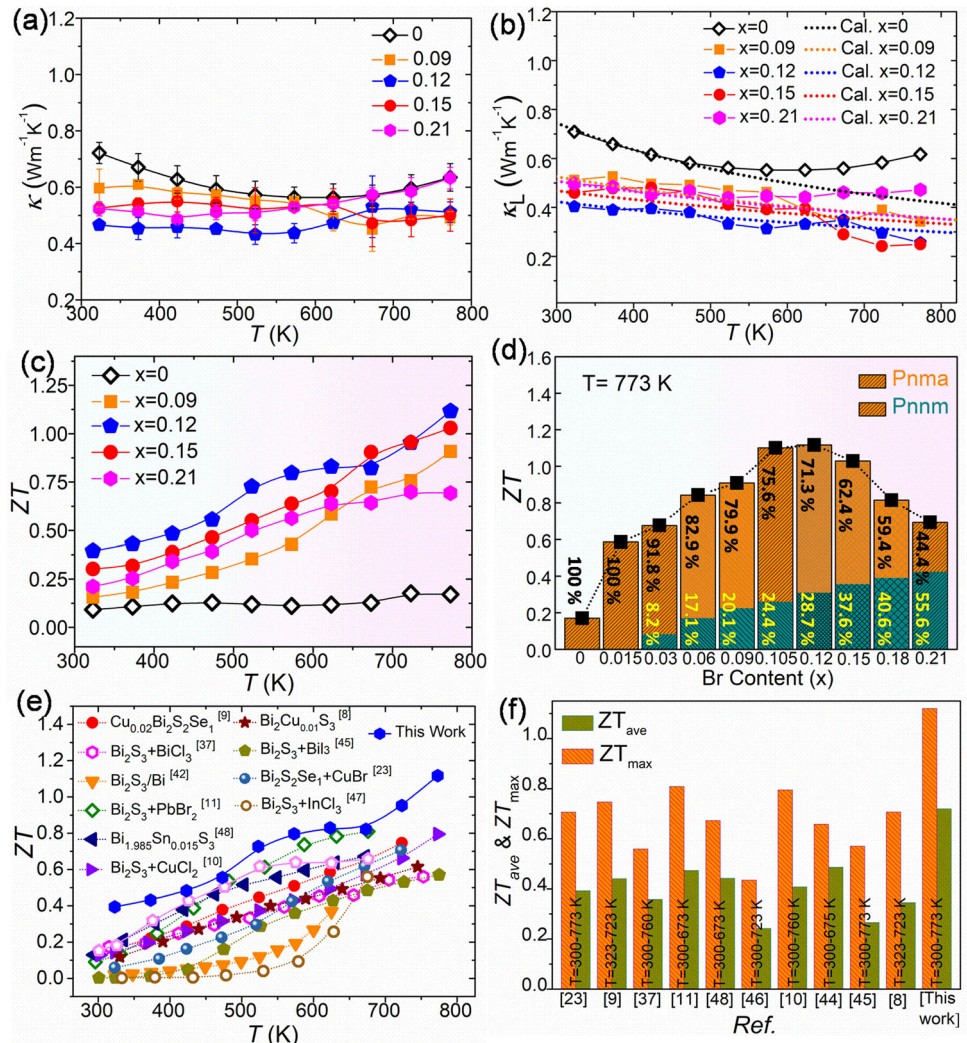

**Fig. 7 Thermal transport properties and _ZT_ values. a** Temperature dependence of the total thermal conductivity ($\kappa$) of $Bi_2Se_{1-x}Br_xS_2$ ($x = 0$, 0.09, 0.12, 0.15, 0.21). Error bars were estimated from the repeatability of the experimental result; three measurements were carried out for each material. **b** Experimental (symbols) and theoretically derived (dotted lines) lattice thermal conductivity ($\kappa_L$) of $Bi_2Se_{1-x}Br_xS_2$ ($x = 0$, 0.09, 0.12, 0.15, 0.21). **c** _ZT_ values of $Bi_2Se_{1-x}Br_xS_2$ ($x = 0$, 0.09, 0.12, 0.15, 0.21). **d** _ZT_ values as a function of Br content, with the bars showing the _Pnma_ and _Pnnm_ phase percentages inside the doped systems. **e-f** Comparison of $ZT_{ave}$ and $ZT_{max}$ values of Br-doped $Bi_2SeS_2$ (this work) with other typical sulfide-based materials reported in the literature[9–11,23,37,42–48].

compounds (Fig. 7(e,f))[8–11,24,37,42–48]. Our work shows that the _ho_C-_he_S nanocomposite of _Pnma_ $Bi_2SeS_2$ - _Pnnm_ $Bi_2SeS_2$ is a promising TE material. The _ho_C-_he_S nanocomposite through a dopant-induced phase transition is an effective strategy to decouple the transport of electrons and phonons and boost the TE figure-of-merit.

## Discussion

This work experimentally showed that the quasi _ho_C-_he_S nanocomposite _Pnma_ $Bi_2SeS_2$ - _Pnnm_ $Bi_2SeS_2$, whose phase composition was induced by a Br dopant, demonstrated excellent TE properties. The properties of this _ho_C-_he_S nanocomposite are very different from those of previously reported nanocomposites with _he_C-_ho_S or _he_C-_he_S interfaces. Br served as an effective carrier donor and also induced a partial $Bi_2SeS_2$ phase transition from _Pnma_ to _Pnnm_, forming the _ho_C-_he_S nanocomposite. The observed Bi–Se elongated bond length clearly predicted the doping-dependent local structural disorder in the doped orthorhombic $Bi_2SeS_2$ system. The coherent interface between the _Pnnm_ nanoprecipitates and _Pnma_ matrix resulted in strongly enhanced

phonon scattering and only slightly impacted the transport of electrons. As a result, a high _PF_ of more than 7.30 $\mu W\ cm^{-1}\ K^{-2}$ at 773 K was obtained, inducing a record high $ZT_{max}$ of 1.12 and a record high $ZT_{ave}$ of 0.72 (at 323–773 K) in the optimal Br-doped _Pnma_ $Bi_2SeS_2$ - _Pnnm_ $Bi_2SeS_2$ nanocomposite (nominal composition: $Bi_2Se_{0.88}Br_{0.12}S_2$). This work provides a general strategy for enhancing TE properties by designing _ho_C-_he_S nanocomposites through a dopant-induced phase transition.

## Methods

Polycrystalline $Bi_2Se_{1-x}Br_xS_2$ (where $x = 0$, 0.015, 0.03, 0.06, 0.09, 0.0105, 0.12, 0.15, 0.18, or 0.21) powders were prepared by melting stoichiometric amounts of high-purity elements (>99.99%) packed under vacuum in glass ampoules at 1173 K for 10 h, followed by annealing at 773 K for 48 h. The obtained ingots were pulverized into fine micron-sized powders by hand grinding. The powders were then compacted at 773 K by spark plasma sintering for 10 min in vacuum under a pressure of 60 MPa.

X-ray diffractometry (XRD) (Riguku, Japan) analysis was performed on all specimens with Cu K$\alpha$ radiation, a wavelength of $\lambda = 1.5406$ Å, and a scanning speed of 4°/min. Rietveld refinement analyses were carried out by using the Generalized Structural Analysis System (GSAS-II) program. Field emission scanning electron microscopy (FE-SEM) (Ultra 55, Zeiss) was utilized to perform microstructure analysis of the freshly fractured surfaces of specimens. High-

resolution transmission electron microscopy (HRTEM) (JEOL-F2010, acceleration voltage of 200 kV) was employed to characterize the microstructures. A ZEM-3 apparatus (ULVAC-Riko) was used to measure the Seebeck coefficient and electrical resistivity in a helium atmosphere from 300 to 550 K. A laser flash method with a commercial system (Netzsch, LFA-427) was used to measure the thermal diffusivity ($D$) in the identical direction (in-plane) in electrical resistivity measurements to avoid overvaluing $ZT$. Differential scanning calorimetry (DSC) (Netzsch, DSC404-C) was employed to determine specific heat ($C_p$). Density ($d$) was determined through the Archimedes method (for all $Bi_2Se_{1-x}Br_xS_2$ samples, the relative density ranges from 98 to 96% as x increases from 0 to 0.21). The thermal conductivity ($\kappa$) of the specimens was obtained by using the equation $\kappa = DdC_p$. Hall coefficients ($r_H$) were evaluated through a physical properties measurement system (PPMS, Quantum design) (Fig. S14). The carrier concentration ($n$) was calculated by $n = 1/(er_H)$, where $e$ is the electronic charge, $r_H$ is the Hall coefficient. The carrier mobility ($\mu$) was calculated by $\mu = r_H/\rho$, where $\rho$ is the electrical resistivity, which was measured from a ZEM-3 apparatus (ZEM-3, ULVAC-Riko). Sound velocity measurements were performed at room temperature on all samples. Longitudinal and transverse sound velocities were determined by using a pulse-receiver (Olympus-NDT) equipped with an oscilloscope (Keysight). The accuracies of the resistivity, Seebeck coefficient, and thermal conductivity measurements are approximately ±2%, ±5%, and ±5%, respectively. The uncertainty of $ZT$ is about ±10%.

DFT calculations were performed using a pseudo-potential projector augmented wave method with the Perdew-Burke-Ernzerh of generalized gradient approximation exchange-correlation potential as implemented in the Vienna Atomic Simulation Package. A $2 \times 2 \times 1$ supercell containing 80 atoms was constructed with its Brillouin zone sampled with a $1 \times 3 \times 2$ k-point mesh. The plane wave cutoff energy was set to 350 eV and the atomic coordinates were relaxed until the total energy converged to $10^{-5}$ eV. The calculated bandgap of pristine $Bi_2SeS_2$ ($Bi_{32}Se_{16}S_{32}$) is 1.14 eV, in good agreement with previously reported values. For Br doping at Se sites, the doped system was obtained by replacing one Se atom with one Br atom (for instance, $Bi_{32}Se_{15}Br_1S_{32}$) as the object of calculations. This corresponded to a doping concentration of ~6.25%. The difference in doping concentration between the experimental and computational experiments is due to the limitation of computational capacity, meaning the lower doping levels reported in the experimental scheme are unable to be computationally investigated. Therefore, these calculation results can only be discussed in a qualitative sense.

**Reporting summary**. Further information on research design is available in the Nature Research Reporting Summary linked to this article.

## Data availability

All data generated or analyzed during this study are included in the published article and its Supporting Information. The data that support the findings of this study are available from the corresponding author (lifu@szu.edu.cn) upon reasonable request.

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

## Acknowledgements

This work was supported by the National Natural Science Foundation of China (No. 52072248), National Key Research and Development Program of China (No. 2018YFB0703600), Natural Science Foundation of Guangdong Province of China (No. 2021A1515012128, No. 2018A030313574), Guangdong Innovative and Entrepreneurial Research Team Program (No. 2016ZT06G587), and Natural Science Foundation of SZU (No. 827-000357), as well as Shenzhen Key Projects of Long-Term Academic Support Plan (No. 20200925164021002). F.L. and B.J. also wish to acknowledge the HAADF-STEM assistance provided by the Electron Microscope Center of Shenzhen University. W.L. acknowledges the Tencent Foundation through the XPLORER PRIZE and the Guangdong Provincial Key Laboratory Program (2021B1212040001) of the Department of Science and Technology of Guangdong Province, China.

## Author contributions

The paper was prepared though the contribution of all authors. B.J., F.L. and W.L. designed the work. B.L., Z.Z., C.L. and D.A. prepared the nanocomposite and measured the thermoelectric transport properties. B.L. and Y.C. performed structrural nano-composite characterization. A.M. and Y.Z. performed simulation. Z.Z. and P.F. planed and supervised the work. F.L. and W.L. wrote the paper. B.L. and G.L. had major input in the wirting of the paper. All the authors edited the paper.

## Competing interests

The authors delcare no competing intrerests.
