## [Peer review file · Nature Communications]

REVIEWER COMMENTS

Reviewer #1 (Remarks to the Author):

This paper reports a significant enhancement of thermoelectric performance in the Bi₂SeS₂ system with Br doping. With Br doping, the system formed a nanocomposite, orthorhombic Pnma matrix with the nanoinclusion of orthorhombic Pnnm phases; however, their chemical compositions are the same. This induced phase transition significantly enhances the electrical conductivity and reduces the thermal conductivity in the Bi₂SeS₂ composite. Consequently, a high figure-of-merit of about 1.2 at 773 K is achieved in the optimal Bi₂S₂Se_{0.88}Br_{0.12} composition. The present work is of interest, and it would contribute to the understanding of the transport of electrons and phonons in homo-composition and hetero-structure nanocomposite PnmaBi₂SeS₂-Pnnm Bi₂SeS₂. However, before the acceptance, I recommend the authors address the comments listed below.

1. The lattice parameter in Br-doped nanocomposite shows no systematic variation with Br-content. Therefore, how the authors are confirmed the Br is successfully replaced the Se in the Bi₂SeS₂ system. Besides, the authors need to explain why Br is chosen to substitute in the Bi₂SeS₂ system.
2. I suggest that the authors should check the XRD data of the undoped Bi₂SeS₂ sample. I found a small peak for the Pnnm phase. I will also found that the colors for the observed and the calculated XRD patterns in Fig. 2(b) are not very clear. Why are the intensities of the peak (011) and (301) changing significantly with Br?
3. How did the authors calculate the carrier concentration and mobility of all samples? Why does the charge carrier concentration increases with Br in Bi₂SeS₂? Is it related to differences in valence electrons, electronegativity, or atomic radius?
4. The authors claimed that the electronic density of states changes due to the activation of conduction bands via doping. In this case, using the Seebeck coefficient data, the authors can evaluate how the Fermi energy and band-gap energy of the presented alloys change with Br.
5. A comparison of the TE properties, such as electrical conductivity and the Seebeck coefficient, with other Bi₂S₃ related systems is required in the manuscript.
6. As the magnitude of electrical conductivity changes considerably with Br doping, therefore, I expect that the electronic thermal conductivity should change with Br content. In this respect, the authors need to show the temperature-dependent electronic thermal conductivity of all samples.
7. For the increase in thermal conductivity at high temperatures, the authors mentioned the bipolar effect in the Bi₂SeS₂ sample. Here, I suggest that the authors calculate the bipolar thermal conductivity contribution using the Seebeck coefficient and electrical conductivity data [Phys. Rev. Appl. 14, 054046 (2020)].
8. In Fig. 7(f), I do not understand the references 1, 2, 3,,10. The authors showed a fraction of Pnma and Pnnm phases in Fig. 7(d), which is not apparent. Also, I do not see the dotted line for the calculated lattice thermal conductivity in Fig. 7(b). How do the authors calculate the average ZT of the present sample?

Reviewer #2 (Remarks to the Author):

In the manuscript, the authors reported the quasi homo-composition and hetero-structure (hoC-heS) nanocomposite consisting of Pnma Bi₂SeS₂- Pnnm Bi₂SeS₂ which was obtained through a Br dopant-induced phase transition. The reported hoC-heS nanocomposite is novel and unique compared with those of previously reported with heC-hoS or heC-heS interfaces. The observed doping-dependent local structural disorder and dopant-induced phase transition resulted in strongly enhanced phonon scattering and only slightly impacted the transport of electrons, and decouples the transport of phonons and electrons. Thus, it not only enhanced the power factor, but also decreased the lattice thermal conductivity. Hence, the Br-doped Pnma Bi₂SeS₂ - Pnnm Bi₂SeS₂ nanocomposite achieved a high ZT of ~1.12 (at 773 K) and an high average ZT of ~0.72 (at T= 323 - 773 K), respectively. The manuscript is well organized and would be a good reference for future research in the

thermoelectricity research and also a good reference for designing nanocomposite. Therefore, I feel that this manuscript is significant enough for publishing in Nature Communication. It can be accepted with some minor revisions as following.

- (1) In supporting information, in Table S1, the units of content x should be corrected.
- (2) The structural analysis show inter-orthorhombic phase transition. We know that it is caused by deviations in lattice parameters due to modifications of the band distances and strain. Is there any possibility for the existence of macrostrain?
- (3) Author described that the strain in ϵ_{xx} and ϵ_{yy} directions show continuous contrast at the interfaces, resulting that the strain is negligible at the interface. It is better to confirm the strain contrast from rotational strain profile at phase boundaries.
- (4) The author related enhancement of power factor with both Seebeck coefficient and electrical conductivity. Can author describe the role of density of state effective mass on Seebeck coefficient?
- (5) Some typos should be corrected: The lattice thermal conductivity should presented by the same term: κ_l should corrected to κ_L . The same term κ_{lat} should be corrected (in the last paragraph of introduction section) to κ_L . Similar mistakes on following pages should be corrected.
- (6) Author should clarify the meaning of the sentence following Fig. 6 "Moreover, the Fermi level is prominently inclined toward the conduction band (CB) in Br-doped Pnma Bi₂SeS₂, indicating that the concentration of electrons is significantly higher than the concentration of holes and that the value of n is also higher".
- (7) I suggest to compare the current results with the reported n-type sulfide-family in the last two years, which will increase the readership.

Reviewer #3 (Remarks to the Author):

The manuscript reports on the impact of Br doping in Bi₂SeS₂, which induces a structural phase transition from Pnma to Pnnm space group for $x > 0.015$. The carrier concentration is increased by the Br substitution, and moreover the presence of two different nanocomposites is reported to reduce the lattice thermal conductivity. Both effects lead to an enhancement of the ZT values. Compared to previous works, the originality comes from the coexistence of two different crystallographic structures in the same material with almost the same chemical composition.

Several state of the art experimental techniques have been used for this work. Moreover, electronic structure calculations and calculations of defects formation energy have been performed to support the hypothesis of Se by Br substitution, and of the role of Br on the electronic gaps. This work is thus interesting and presents a new and elegant approach to optimize ZT values in chalcogenides. There are several points which need to be clarified in the manuscript, points which would deserve a more detailed discussion.

- Line 118 : is the decrease of lattice parameters really significant ? What would be expected from a Vegard's law ?
- The description of Formation Energy in the Supplementary part should be described in more details to justify the use of the chemical formula. And is this formulation coming only from the formation energy calculations or is there some experimental proof of the Br location?
- Graph 5a : why are the x values for $x = 0.12$ and $x = 0.21$ shifted on the x values ?
- As shown in S7, the Pisarenko plot fails for x values larger than 0.015. Does it have any impact on the Hall effect curves ? Only the n values are given in Fig.5a, but can the authors comment on the extraction of n values by themselves ? Is it still a single band model, with a linear behavior of $R(H)$? These curves should be given in the Supplementary data.
- Line 224 : In Figure 7b, it is impossible to read the Callaway fits. The figure caption may be wrong as

there are no dotted lines, it rather seems to be straight lines. Also the same symbols should be used for all the curves, and the same as in the supplementary curves (see for example $x=0$ in Fig. 7a and 7b, two different symbols are used).

- Lines 234 : the authors should explain why the parameters B and C in the Callaway fits are chosen to remain constant, and how these B and C values were chosen ?

Minor typos

Line 77 : 'very different from those'

Line 95 : 'predicted structure'

Line 105 : 'planar transitions'

Line 169 : 'the magnitude of S' rather than 'S'

Supplementary : 'energy dependent constant)'

Line 253 : 'different from'

To conclude, the effect of 'homo-composition and hetero-structure' seems to be new and interesting to improve the ZT values of these chalcogenides. The paper is well organized but more details should be given on the description of transport data, to give a more precise description of the fitting parameters and on the impact of 'hetero-structuring' on the electronic transport.

Dear Reviewers,

We would like to thank you very much for your comments and suggestions which were extremely helpful to further improve our manuscript (NCOMMS-21-33758). We have taken on board all these comments and revised the manuscript very closely to what was advised. Please find below our detailed reply to each comment as well as our revised manuscript, where the modified parts are marked blue. We hope the revisions are acceptable.

Thank you very much in advance.

Revised details:

Reviewer #1 (Remarks to the Author): *This paper reports a significant enhancement of thermoelectric performance in the Bi_2SeS_2 system with Br doping. With Br doping, the system formed a nanocomposite, orthorhombic $Pnma$ matrix with the nanoinclusion of orthorhombic $Pnmm$ phases; however, their chemical compositions are the same. This induced phase transition significantly enhances the electrical conductivity and reduces the thermal conductivity in the Bi_2SeS_2 composite. Consequently, a high figure-of-merit of about 1.2 at 773 K is achieved in the optimal $\text{Bi}_2\text{S}_2\text{Se}_{0.88}\text{Br}_{0.12}$ composition. The present work is of interest, and it would contribute to the understanding of the transport of electrons and phonons in homo-composition and hetero-structure nanocomposite $Pnma \text{ Bi}_2\text{SeS}_2 - Pnmm \text{ Bi}_2\text{SeS}_2$. However, before the acceptance, I recommend the authors address the comments listed below.*

1. The lattice parameter in Br-doped nanocomposite shows no systematic variation with Br-content. Therefore, how the authors are confirmed the Br is successfully replaced the Se in the Bi_2SeS_2 system. Besides, the authors need to explain why Br is chosen to substitute in the Bi_2SeS_2 system.

Reply: Thank you. We have two pieces of evidences for Br replacing the Se in the Bi_2SeS_2 : (1) the systematic variation of the ratio between $Pnma$ phase and $Pnmm$ phase, and (2) the systematic variation of carrier concentration. Regarding evidence (1), the molar fraction based on the XRD pattern could be a piece of good evidence

for the dopant effect of Br. For evidence (2), Br is a well-known n-type dopant for the Bi_2X_3 family (X=Te, Se, S), with a defect Br_x providing a free electron. In the Br doped Bi_2SeS_2 , we have observed this increased carrier concentration in the *Pnma* phase dominant regions. To furthermore confirm the Br would likely get into the Se sub-lattice rather than the S sub-lattice, we also conducted a formation energy calculation. The calculated result in Fig. S11 reveals that the formation energy values for Br doped at Se site are lower than that at S site, indicating the Br prefers to occupy the site of Se. Thus, combined with the experiment data and the formation energy calculation, it is confirmed that Br has successfully replaced the Se in the Bi_2SeS_2 system. To make it much clear, we have added a sentence in the revised manuscript as follows.

Added: "Besides, the calculated formation energy also shows that the formation energy value for Br doping at Se site is much lower than that at S site (Fig. S11(a)). This means Br prefers to occupy the Se site in Bi_2SeS_2 ."

Regarding “*no systematic variation in lattice parameter*”, it could be XRD refinement accuracy limit for the lattice parameter, in our facility, due to the co-existence of two phases. However, it didn’t change our main conclusion.

Additionally, as mentioned above, halogen elements (Cl, Br, I) are well-known n-type dopant for the Bi_2X_3 family (X=Te, Se, S), with a defect Cl_x , Br_x , or I_x providing a free electron to increase the carrier concentration. For example, the carrier concentration for the undoped Bi_2S_3 is only $3.7 \times 10^{16} \text{ cm}^{-3}$. However, it significantly increased to 1.0×10^{19} and $2.6 \times 10^{19} \text{ cm}^{-3}$ after doping 0.25mol% BiCl_3 and 0.5mol% BiCl_3 (*Adv. Energy Mater.* **2**, 634-638(2012)). Meanwhile, after doping 0.5% CuBr_2 and 1.5mol% PbBr_2 , the carrier concentration can be improved to $5.6 \times 10^{19} \text{ cm}^{-3}$ and $4.8 \times 10^{19} \text{ cm}^{-3}$ (*Nano Energy* **13**, 554(2015); *Nano Energy* **78**, 105227(2020)), which can optimize the electrical transport property of the sample. For the same purpose in the present study, Br was chosen to substitute in the Bi_2SeS_2 system. As described in the manuscript, the carrier concentration of the undoped Bi_2SeS_2 with $1.28 \times 10^{19} \text{ cm}^{-3}$ has obviously increased to $4.38 \times 10^{20} \text{ cm}^{-3}$ after doping 3% Br. However, we also find

the phase transitions due to Br doping in Bi_2SeS_2 system, which is very helpful to reduce the lattice thermal conductivity with negligible impact on the electrical transport property. In order to make it much clear, we have added a sentence in the Introduction section in the revised manuscript as follows.

Added: "Br element is familiarly donor in the Bi_2S_2 , Bi_2Se_3 , and Bi_2SeS_2 ."

2. I suggest that the authors should check the XRD data of the undoped Bi_2SeS_2 sample. I found a small peak for the $Pnmm$ phase. I will also found that the colors for the observed and the calculated XRD patterns in Fig. 2(b) are not very clear. Why are the intensities of the peak (011) and (301) changing significantly with Br?

Reply: Thank you. (1) We have re-draw the Fig. 2(b) to clarify the observed and the calculated XRD patterns. (2) We have checked the XRD data of all the samples. Some main characteristic peaks of the XRD patterns have been enlarged to compare with the standard peaks of $Pnmm$ phase as shown in the following Fig. R1. We didn't find the peaks for the $Pnmm$ phase in the Br-free sample. It is clear that the $Pnmm$ phase appears with increasing Br content x . With further increasing the Br content x , the peaks gradually shifted toward the higher 2θ . To clarify, we have added the figure (Fig. R1) as follows in the revised supporting information as Fig. S2. In addition, the calculated standard XRD peaks for $Pnma$ and $Pnmm$ phases were also added in Fig. 2(a) to make it much clear.

Fig. R1 Some main characteristic peaks of the XRD patterns for all the samples

(3) The intensities of the peaks (011) and (301) for the *Pnma* phase changed with Br doping, indicating continuous lattice distortion before the transition into the *Pnmm* phase, which has confirmed by the HRTEM as described in the manuscript. In addition, it is reported that the stability of various surfaces can be quantitatively described by their surface energies (*J Chem. Phys.* **143**, 151101 (2015)). Thus, in order to find the doping-induced lattice distortion inside *Pnma* structure, DFT calculations were performed along (011) & (301) plane, which was presented in Fig. S11(b) in the revised supporting information. It is found that the formation energy along the plane (011) has reduced, while the formation energy along the plane (301) has increased after Br doping. This means the plane (011) might be easy to form during the preparation due to the low energy after Br doping, which could possibly cause the changing intensities of the peaks (011) and (301) with Br doping.

To make it clear to the readers, we have added more descriptions in the revised manuscript as follows.

Added: "In addition, it is reported that their surface energies can quantitatively describe the stability of various surfaces.⁴⁰ The calculated formation energy along the plane (011) and (301) indicates that the formation energy along the plane (011) has reduced, while it has increased along the plane (301) after Br doping. This means the plane (011) might be easy to form during the preparation due to the low energy after Br doping."

3. *How did the authors calculate the carrier concentration and mobility of all samples? Why does the charge carrier concentration increases with Br in Bi₂SeS₂? Is it related to differences in valence electrons, electronegativity, or atomic radius?*

Reply: (1) The Hall coefficients, r_H , of all the samples were measured at room temperature using a physical properties measurement system (PPMS, Quantum design). The carrier concentration (n) was calculated by $n=1/(er_H)$, where e is the electronic charge, r_H is the Hall coefficient. The carrier mobility (μ) was calculated

by $\mu=r_H/\rho$, where ρ is the electrical resistivity, which was measured from a ZEM-3 apparatus (ZEM-3, ULVAC-Riko). In order to make it much clear, this information was added in the Experimental section as follows.

Added: "Hall coefficients (r_H) were evaluated through a physical properties measurement system (PPMS, Quantum Design) (Fig. S10). The carrier concentration (n) was calculated by $n=1/(er_H)$, where e is the electronic charge, r_H is the Hall coefficient. The carrier mobility (μ) was calculated by $\mu=r_H/\rho$, where ρ is the electrical resistivity, which was measured from a ZEM-3 apparatus (ZEM-3, ULVAC-Riko)."

(2) Mainly, the Br atom has one more valence electron than the Se atom. Thus, the substitution of Se with Br would offer more extra electrons in the Bi_2SeS_2 matrix according to the defect chemistry reaction as follows, and then enhance carrier concentration for all the doped samples.

Thus, the increase in charge carrier concentration is mainly related to the differences in valence electrons of Br and Se. This is also the main reason that why Br is chosen to substitute in the Bi_2SeS_2 compound, which has been described in our response for your Comment 1#.

4. *The authors claimed that the electronic density of states changes due to the activation of conduction bands via doping. In this case, using the Seebeck coefficient data, the authors can evaluate how the Fermi energy and band-gap energy of the presented alloys change with Br.*

Reply: Thank you. According to the Goldsmid-Sharp relation as follows (*Physical Review Applied* **14**, 054046(2020); *Journal of Electronic Materials* **28**, 869(1999)),

$$S_{max}=E_g/2eT_{max} \quad (1)$$

where e is the elementary charge, the Seebeck band gap E_g has been estimated by the maxima in $S(T)$ with the coordinates (T_{max}, S_{max}) . Moreover, the reduced Fermi energy at room temperature can be estimated via Seebeck coefficient (S) by using the

following equations,

$$S = \frac{k_B}{e} \left[\frac{(\lambda+2)F_{\lambda+1}(\xi_F)}{(\lambda+1)F_{\lambda}(\xi_F)} - \xi_F \right] \quad (2)$$

$$m_d^* = \frac{h^2}{2k_B T} \left(\frac{n}{4\pi F_{1/2}(\xi_F)} \right)^{2/3} \quad (3)$$

where $F_j(\xi_F) = \int_0^{\infty} \frac{x^j}{1+e^{x-\xi_F}} dx$ is the Fermi integral of order j , S is the Seebeck coefficient, m_d^* is the density of state effective mass, ξ_F is reduced Fermi energy of $E_F/k_B T$, k_B is the Boltzmann constant and h is Plank constant. Table R1 as following shows the Seebeck band gap (E_g) and Fermi energy (E_F) for all the samples. It can be found that the band gap has been roughly reduced after Br doping. As for the Fermi energy, it firstly increases after doping and then reduces with further increasing the doped content x , indicating the change of the electronic density of states. However, the data is estimated in the case of assuming a single parabolic band model. More accurate data might need to be obtained by more complex band model.

In order to well understand the present data, Table R1 as follows was added in the revised supporting information as Table S2. The calculation method above was also given in the supporting information.

NOTE: Eq. (1-3) above are derived from the assumption of single uniform phase. In our case, we have two phases, which may result into the physical meaning of E_g is not very clear.

Table R1 Seebeck coefficient (S), estimated Seebeck band gap (E_g) and Fermi energy (E_F) at room temperature for all the samples $\text{Bi}_2\text{Se}_{1-x}\text{Br}_x\text{S}_2$ (x : 0 - 0.21).

x	0	0.015	0.03	0.06	0.09	0.105	0.12	0.15	0.18	0.21
$S(\mu\text{V/K})$	297.2	58.2	58.9	92.5	139.3	231.4	214.1	197.3	218.3	257.8
$E_g(\text{eV})$	0.47	0.27	0.26	0.28	0.32	0.24	0.2	0.21	0.21	0.23
$E_F(\text{eV})$	-0.035	0.124	0.122	0.071	0.033	-0.0036	-0.0042	0.0031	-0.0057	-0.021

5. A comparison of the TE properties, such as electrical conductivity and the Seebeck coefficient, with other Bi_2S_3 related systems is required in the manuscript.

Reply: Thank you for your suggestion. We have listed the thermoelectric

properties, including electrical conductivity, Seebeck coefficient, thermal conductivity and ZT value, of some representative Bi_2S_3 related systems in a table as follows (Table R2). The present data for $\text{Bi}_2\text{Se}_{1-x}\text{Br}_x\text{S}_2$ ($x=0.12$) in our work was also presented in the table to show the comparison. We can find that $\text{Bi}_2\text{Se}_{1-x}\text{Br}_x\text{S}_2$ possesses moderate Seebeck coefficient but relatively higher electrical conductivity and lower thermal conductivity among all the materials listed in the table, which benefitting from the significantly reduced lattice thermal conductivity while the negligible impact on the electrical transport property by the *hoC-heS* nanostructure in our work. Thus, it shows clearly that the ZT value of $\text{Bi}_2\text{Se}_{1-x}\text{Br}_x\text{S}_2$ with 1.12 at 773 K is apparently higher as compared with other Bi_2S_3 related compounds.

In the revised supporting information, we have added the table (Table R2 in the following) as Table S4. And all the references were added and reorganized orderly. In order to make it much clear to the readers, we also added more discussion in the revised manuscript (main text) as follows.

Added: "As a comparison, Table S4 and Fig. S14 present the TE properties, including σ , S , κ , and ZT value, of some representative Bi_2S_3 related systems. Both the ZT_{\max} (1.12 at 773 K) and ZT_{ave} (0.72 in 323 – 773 K) values of as-fabricated *hoC-heS* nanocomposite of *Pnma* Bi_2SeS_2 - *Pnnm* Bi_2SeS_2 are much higher than previously reported Bi_2SeS_2 and Bi_2S_3 -based materials, and other reported sulfide compounds (Fig. 7(e-f)).^{8-11,24,37,42-48} Our work shows that the *hoC-heS* nanocomposite of *Pnma* Bi_2SeS_2 - *Pnnm* Bi_2SeS_2 is a promising TE material. The *hoC-heS* nanocomposite through a dopant-induced phase transition is an effective strategy to decouple the transport of electrons and phonons and boost the TE figure-of-merit."

Table R2 Thermoelectric properties, including electrical conductivity (σ), Seebeck coefficient (S), thermal conductivity (κ) and ZT_{\max} value at the optimum temperature (T_{\max}) of the present $\text{Bi}_2\text{Se}_{1-x}\text{Br}_x\text{S}_2$ ($x=0.12$) and other reported Bi_2S_3 related compounds.

Sample	$\sigma(\text{Scm}^{-1})$	$S(\mu\text{VK}^{-1})$	$\kappa(\text{Wm}^{-1}\text{K}^{-1})$	ZT_{\max}	$T_{\max}(\text{K})$
$\text{Bi}_2\text{Se}_{1-x}\text{Br}_x\text{S}_2$ ($x=0.12$)	~210	187	0.51	1.12	773(this work)
$\text{Bi}_2\text{S}_3+1.5\text{mol}\%\text{PbBr}_3$	~90	260	0.50	~0.8	673 ¹

$\text{Bi}_{2-x}\text{Sn}_x\text{S}_3$ ($x=0.015$)	~65	~-190	~-0.22	0.67	673^2
$\text{Cu}_{0.01}\text{Bi}_2\text{S}_3$	~67	~-250	~-0.5	0.62	723^3
$\text{Cu}_{0.02}\text{Bi}_2\text{SeS}_2$	146	~-200	~-0.36	0.75	723^4
$\text{Bi}_2\text{S}_3+0.5\text{mol}\% \text{CuCl}_2$	~110	~-220	~-0.55	~0.8	760^5
$\text{Bi}_2\text{S}_3+0.5\text{mol}\% \text{BiCl}_3$	107	-233	~-0.80	0.6	760^6
$\text{Bi}_2\text{S}_3/\text{Bi}$	~10	~-400	~-0.36	0.36	623^7
Bi_2S_3	~200	~-150	~-0.65	0.5	723^8
$\text{Bi}_2\text{S}_3+1.0 \text{ mol}\% \text{ZnO}$	~75	~-250	~-0.47	0.66	675^9
$\text{Bi}_2\text{S}_3+1.0 \text{ mol}\% \text{BiI}_3$	22	~-370	0.42	0.58	773^{10}
$\text{Bi}_2\text{SeS}_2+3\text{mol}\% \text{CuBr}$	~140	~-200	~-0.55	0.71	723^{11}
$\text{Bi}_2\text{S}_3+1 \text{ mol}\% \text{InCl}_3$	62	-244	0.42	0.57	673^{12}
$\text{Bi}_2\text{SeS}_2+x\text{CuI}$ ($x=0.02$)	~129	220	0.47	1.04	773^{13}

References

1. Guo, J., et al. *Nano Energy* **78**,105227 (2020).
2. Guo, Y., Du, X. L., Wang, Y. L. & Yuan, Z. H. *J. Alloys Compd.* **717**, 177-182 (2017).
3. Yang, J. et al. *J. Alloys Compd.* **780**, 35-40 (2019).
4. Li, L. et al. *Nano Energy* **12**, 447-456 (2015).
5. Ji, W. T. et al. *Nano Energy* 106171 (2021).
6. Biswas, K., Zhao, L. -D. & Kanatzidis, M. G. *Adv. Energy Mater.* **2**, 634-638 (2012).
7. Ge, Z. H. et al. *ACS Appl. Mater. Inter.* **29**, 4828-4834 (2017).
8. Liu, W. S. et al. *Nano Energy* **4**, 113-122 (2014).
9. Du, X. et al. *RSC Adv.* **5**, 31004-31009 (2015).
10. Yang, J. et al. *J. Alloys Compd.* **728**, 351-356 (2017).
11. Ruan, M., Li, F., Chen, Y. X., Zheng, Z. H. & Fan, P. *J. Alloys Compd.* **849**, 156677 (2020).
12. Guo, J., Ge, Z. H., Qian, F., Lu, D. H. & Feng, J. *J. Mater. Sci.* **55**, 263-273 (2020).
13. Li, F., et al. *Nano Energy* **88**, 106273 (2020).

6. As the magnitude of electrical conductivity changes considerably with Br doping, therefore, I expect that the electronic thermal conductivity should change with Br content. In this respect, the authors need to show the temperature-dependent electronic thermal conductivity of all samples.

Reply: We have calculated the electronic thermal conductivity (κ_e) from the equation of $\kappa_e=L\sigma T$, where σ is the electrical conductivity, T is the temperature, L is the Lorenz number obtained by applying the calculated reduced Fermi energy η and scattering parameter r , ranging from 1.5×10^{-8} to 2.0×10^{-8} V^2/K^2 . The temperature-dependent κ_e of all the samples $\text{Bi}_2\text{Se}_{1-x}\text{Br}_x\text{S}_2$ is shown in Fig. R2 as follows, which presents good consistency with the electrical conductivity σ . Compared with the undoped sample, the κ_e increases firstly after doping due to the

enhanced electrical conductivity and holds ~36% of the total thermal conductivity for the doped sample with $x=0.015$. And then it reduces because of the decrease of the electrical conductivity and contributes ~18% of the total thermal conductivity for the sample with doped content $x=0.09$. Thus, it suggests that the lattice thermal conductivity dominates the total thermal conductivity. This figure was added in the revised supporting information as shown in Fig. S12. Some discussions were also added in the supporting information. Details are as follows.

Added: "The electronic thermal conductivity (κ_e) was calculated from the equation of $\kappa_e=L\sigma T$, where σ is the electrical conductivity, T is the temperature, L is the Lorenz number obtained by applying the calculated reduced Fermi energy η and scattering parameter r , ranging from 1.5×10^{-8} to 2.0×10^{-8} V^2/K^2 . The temperature-dependent κ_e of all the samples $\text{Bi}_2\text{Se}_{1-x}\text{Br}_x\text{S}_2$ is shown in Fig. S12 (b), which presents good consistency with the electrical conductivity σ . Compared with the undoped sample, the κ_e increases firstly after Br doping due to the enhanced electrical conductivity and holds ~36% of κ for the doped sample with $x=0.015$. And then it reduces because of the decrease of the electrical conductivity and contributes ~18% of κ for the sample with doped content $x=0.09$. Thus, it suggests that the lattice thermal conductivity dominates the total thermal conductivity."

Fig. R2 Temperature dependence of the electronic thermal conductivity (κ_e) of all Br doped $\text{Bi}_2\text{Se}_{1-x}\text{Br}_x\text{S}_2$ samples ($x = 0, 0.015, 0.03, 0.06, 0.09, 0.105, 0.12, 0.15, 0.18, 0.21$).

7. For the increase in thermal conductivity at high temperatures, the authors mentioned the bipolar effect in the Bi_2SeS_2 sample. Here, I suggest that the authors calculate the bipolar thermal conductivity contribution using the Seebeck coefficient and electrical conductivity data [*Phys. Rev. Appl.* 14, 054046 (2020)].

Reply: Thank you. Following your suggestions, we have calculated the bipolar thermal conductivity (κ_b) by using the following formula according to the reference (*Phys. Rev. Appl.* 14, 054046 (2020)),

$$\kappa_b = \frac{b}{b+1} \left(\frac{E_g}{k_B T} + 4 \right)^2 \left(\frac{k_B}{T} \right) \sigma T$$

where b is the mobility ratio of the charge carriers, k_B is the boltzmann constant, σ is the measured electrical conductivity, and E_g is the intrinsic energy gap which derived from the equation $S_{\max} = E_g / 2eT_{\max}$ by the maxima Seebeck coefficient (S) in $S(T)$ with the coordinates (T_{\max}, S_{\max}) as seen in Table R1 above. However, the calculated bipolar thermal conductivity is over-estimated, even higher than the total thermal conductivity, which might be due to parameter b ($b=1$) and the estimated energy gap used in the calculation.

In order to clarify the contribution of κ_b at high temperatures, the κ_b is separated from the total thermal conductivity (κ) according to the method proposed by Kitagawa et al. (*J. Phys. Chem. Solids* 66, 1635 (2005)). The total thermal conductivity (κ) can be written in the form: $\kappa = \kappa_L + \kappa_e + \kappa_b$, where κ_L is the lattice thermal conductivity and κ_e is the electronic thermal conductivity. The difference, $\kappa_L + \kappa_b$ as a function of T^{-1} for the samples $\text{Bi}_2\text{S}_2\text{Se}_{1-x}\text{Br}_x$ is shown in Fig. R3(a) as follows. Since the acoustic phonon scattering is predominant at low temperature, the $\kappa_L + \kappa_b$ equals to κ_L , which is proportional to T^{-1} . With increasing the temperature, $\kappa_L + \kappa_b$ started to gradually deviate from such a linear relationship between κ_L and T^{-1} . This implies that the bipolar diffusion starts to contribute to the thermal conductivity. The κ_L at high temperature was estimated by extrapolating the linear relationship between κ_L and T^{-1} , as indicated by the dotted line in Fig. 3R(a) as follows. Hence, κ_b at high temperature should be equal to $\kappa - \kappa_L - \kappa_e$. Fig. R3(b) as follows shows the temperature dependence of the κ_b for

some representative samples. It can be seen that with increasing Br content x from 0 to 0.12, κ_b shows decreasing trend mainly due to the trapping of minority carriers within the intensive nanoscale regions inside the doped samples. However, κ_b increases with further increasing the doping $x \geq 0.15$ which can be ascribed to the extra carrier regeneration/recombination with the unconfined minority carriers due to agglomerated nanophase or reduced nanoscale regions.

In order to make it much clear, we have added the figure as follows (Fig. R3) in the revised supporting information as shown in Fig. S12. The method to estimate the κ_b and the discussion for the change of the κ_b as mentioned above were all added in the revised supporting information.

Fig. R3 Bipolar thermal conductivity (κ_b) for some representative samples $\text{Bi}_2\text{Se}_{1-x}\text{Br}_x\text{S}_2$ ($x=0, 0.015, 0.06, 0.105, 0.12, 0.21$). The dotted line in (a) is linearly fitting to the lattice thermal conductivity (κ_L) at low temperature.

8. In Fig. 7(f), I do not understand the references 1, 2, 3,,10. The authors showed a fraction of $Pnma$ and $Pnmm$ phases in Fig. 7(d), which is not apparent. Also, I do not see the dotted line for the calculated lattice thermal conductivity in Fig. 7(b). How do the authors calculate the average ZT of the present sample?

Reply: Thank you. (1) We have corrected the name of the references in Fig. 7(f). Moreover, in Fig. 7(d), we have marked the fraction of $Pnma$ and $Pnmm$ phase to make it much clear. Also, in Fig. 7(b), the dotted line seems to be straight lines because the dots are too dense. We have corrected the format for lines of theoretically calculated κ_L

values to make it clear.

(2) The average ZT (ZT_{ave}) of the present samples is calculated by integrating the area under the ZT curves according to the following formula,

$$ZT_{ave} = \frac{1}{T_h - T_c} \int_{T_c}^{T_h} ZT dT$$

where $T_c = 323$ K and $T_h = 773$ K are the hot-side and cold-side temperatures.

In order to make it much clear to the readers, the method to calculate the average ZT was added into the revised manuscript. Details are as follows.

Added: "The average ZT_{ave} is calculated by integrating the area under the ZT curves in Fig. 7(c) according to the following formula,

$$ZT_{ave} = \frac{1}{T_h - T_c} \int_{T_c}^{T_h} ZT dT \quad (1)$$

where $T_c = 323$ K and $T_h = 773$ K are the hot-side and cold-side temperatures."

Reviewer #2 (Remarks to the Author): *In the manuscript, the authors reported the quasi homo-composition and hetero-structure (hoC-heS) nanocomposite consisting of Pnma Bi₂SeS₂- Pnnm Bi₂SeS₂ which was obtained through a Br dopant-induced phase transition. The reported hoC-heS nanocomposite is novel and unique compared with those of previously reported with heC-hoS or heC-heS interfaces. The observed doping-dependent local structural disorder and dopant-induced phase transition resulted in strongly enhanced phonon scattering and only slightly impacted the transport of electrons, and decouples the transport of phonons and electrons. Thus, it not only enhanced the power factor, but also decreased the lattice thermal conductivity. Hence, the Br-doped Pnma Bi₂SeS₂ - Pnnm Bi₂SeS₂ nanocomposite achieved a high ZT of ~1.12 (at 773 K) and an high average ZT of ~0.72 (at T= 323 - 773 K), respectively. The manuscript is well organized and would be a good reference for future research in the thermoelectricity research and also a good reference for designing nanocomposite. Therefore, I feel that this manuscript is significant enough*

for publishing in *Nature Communication*. It can be accepted with some minor revisions as following.

1. In supporting information, in Table S1, the units of content x should be corrected.

Reply: Thank you. We have corrected it.

2. The structural analysis show inter-orthorhombic phase transition. We know that it is caused by deviations in lattice parameters due to modifications of the band distances and strain. Is there any possibility for the existence of macrostrain?

Reply: Thanks for the comment. Here, deviations in the lattice parameters of the Br-doped Bi_2SeS_2 indicate the existence of lattice distortion with disordered bond length as presented in Fig. S7 in supporting information and suggest that Br elongates the orthorhombic layered structure as mentioned in Fig. 3 in the manuscript. Thus, this structural distortion (lattice deviation) causes the microstrain in the lattice. We have performed the Geometric phase analysis to confirm the presence of strain inside the present *hoC-heS* system, as presented in Fig. 4 and Fig. S8.

Regarding “*possibility for the existence of macrostrain*”, it could be existence. However, we didn’t have the direct evidence to recognize it. As the macrostrain is large enough, it usually causes a crack in the fast sintered sample. In our case, we didn’t directly observe this kind of crack or microcracks. We conclude that the macrostrain should not be the primary mechanism for the observed thermoelectric transport properties.

3. Author described that the strain in ε_{xx} and ε_{yy} directions show continuous contrast at the interfaces, resulting that the strain is negligible at the interface. It is better to confirm the strain contrast from rotational strain profile at phase boundaries.

Reply: Thank you for your suggestion. The rotation profile (lattice rotation profile) from Geometric phase analysis is further derived to find the strain at phase boundaries as shown in Fig. R4 (d) in following. We find that the rotational strain profile also shows continuous contrast (color) across the phase boundaries. This continuous

contrast (color) across phase boundaries means that the differences of rotation strain at the interfaces are negligible, suggesting that corresponding boundaries are coherent.

In order to make it much clear, we have added a sentence in the revised manuscript. Details are as follows. Fig. R4 as following was also added in the revised supporting information (Fig. S8).

Added: "The rotational strain profile (ϵ_{rot}) also shows continuous contrast across the phase boundaries (Fig. S8)."

Fig. R4 Geometric phase analysis (GPA) of corresponding HRTEM images from Fig. 4 (c) in the main text; (b)-(d) Corresponding strain maps along the ϵ_{xx} , ϵ_{yy} and ϵ_{rot} for (a).

4. The author related enhancement of power factor with both Seebeck coefficient and electrical conductivity. Can author describe the role of density of state effective mass on Seebeck coefficient?

Reply: Thank you. In order to determine whether the density of state effective mass (m_d^*) works and contributes to the Seebeck coefficient in this system, density of state effective mass (m_d^*) are obtained from the single parabolic band model using the following equations, assuming the current composition is single uniform phase:[ACS Nano15(6), 10532-10541 (2021).]

$$S = \frac{k_B}{e} \left[\frac{(\lambda+2)F_{\lambda+1}(\xi_F)}{(\lambda+1)F_{\lambda}(\xi_F)} - \xi_F \right] (1)$$

$$m_d^* = \frac{h^2}{2k_B T} \left(\frac{n}{4\pi F_{1/2}(\xi_F)} \right)^{2/3} (2)$$

where $F_j(\xi_F) = \int_0^{\infty} \frac{x^j}{1+e^{x-\xi_F}} dx$ is the Fermi integral of order j , S is the Seebeck coefficient, m_d^* is the density of state effective mass, ξ_F is reduced Fermi energy of

$E_F/k_B T$, k_B is the Boltzmann constant and h is Plank constant. Fig. R5 as follows demonstrates that as x increases from 0 to 0.12 for $\text{Bi}_2\text{Se}_{1-x}\text{Br}_x\text{S}_2$, the density of state (DOS) effective mass generally increases from $1.62m_e$ ($x=0$) to $6.90m_e$ (at $x=0.21$). The variations in m_d^* values with Br content could be resulted from the band structure change and the interface effect.

Fig. R5(b) as follows and the calculation method of effective mass m_d^* were added in the revised supporting information. In order to make it much clear, some discussion about the effective mass m_d^* and Seebeck coefficient were also added in the revised manuscript. Details are as follows.

Added: "In addition, the density of state (DOS) effective mass (m_d^*) was obtained using the equations (S2)-(S4) in the supporting information, assuming the as-fabricated nanocomposite as quasi-single uniform phase (Fig. S9). A increased m_d^* from $1.62m_e$ ($x=0$) to $6.90m_e$ (at $x=0.21$) with Br content of $\text{Bi}_2\text{Se}_{1-x}\text{Br}_x\text{S}_2$ was observed. The variations in m_d^* values with Br content is resulted from the band structure change and the interface effect."

Fig. R5 (a) Seebeck coefficient (S) as a function of carrier concentration (n), and (b) variation of density of state effective mass with doping content x (at 300 K).

5. Some typos should be corrected: The lattice thermal conductivity should presented by the same term: κl should corrected to κL . The same term $\kappa l a t$ should be corrected (in the last paragraph of introduction section) to κL . Similar mistakes on following pages should be corrected.

Reply: Thank you for your correction. We have corrected all these mistakes.

6. *Author should clarify the meaning of the sentence following Fig. 6 “Moreover, the Fermi level is prominently inclined toward the conduction band (CB) in Br-doped Pnma Bi₂SeS₂, indicating that the concentration of electrons is significantly higher than the concentration of holes and that the value of n is also higher”.*

Reply: Our experimental results have shown that the Br is an effective donor in as-fabricated Bi₂SeS₂ nanocomposite. We also conducted the DFT calculations for the Bi₂Se_{1-x}Se_xS₂, which shows a significant shifting of Fermi level toward the conduction band (CB). This further confirmed the Br provide extra free electrons, and raise the concentration of electrons.

In a semiconductor, the electron and hole concentrations should obey the relationship (*J. Appl. Phys.* **102**, 103717 (2007)):

$$np = N_c N_v e^{(-E_g/k_B T)}$$

where the n , p are the electron and hole concentrations, respectively; N_c , N_v are the effective energy and hole state density of conduction and valence bands, respectively; and E_g is the forbidden gap energy. As a result, the electron concentration should increase with increasing Br while holes decrease at a given temperature.

In order to make it much clear, we have modified this sentence as follows.

Added: "Moreover, the Fermi level is prominently inclined toward the conduction band (CB) in Br-doped Pnma Bi₂SeS₂, confirming the Br is an effective donor. According to the relationship between the electron and hole concentrations and band gap,⁴¹ Br dopant increases the electron concentration but decreases the hole concentration."

7. *I suggest to compare the current results with the reported n-type sulfide-family in the last two years, which will increase the readership.*

Reply: Thank you for your suggestion. We have compared the current results with the reported n-type sulfide-family in the last two years, as given below in Fig. R6. Usually, the thermoelectric property for n-type sulfide compounds with ZT value

lower than 0.8 is much worse than that of the p-type ones because of the lower electrical conductivity and higher thermal conductivity. From Fig. R6 as follows, we can find that most ZT values for these sulfides are lower than 1.0. It is clear that the ZT value of $\text{Bi}_2\text{Se}_{1-x}\text{Br}_x\text{S}_2$ with over 1.0 at 773 K is apparently higher as compared with the other n-type sulfides benefitting from the significantly reduced lattice thermal conductivity while the negligible impact on the electrical transport property by the *hoC-heS* nanostructure in our work.

We have added some sentences in the revised manuscript. Details are as follows. Fig. R6 as following was also added in the revised supporting information (Fig. S14). And all the references were added and reorganized orderly.

Added: "As a comparison, Table S4 and Fig. S14 present the TE properties, including σ , S , κ , and ZT value, of some representative Bi_2S_3 related systems. Both the ZT_{\max} (1.12 at 773 K) and ZT_{ave} (0.72 in 323 – 773 K) values of as-fabricated *hoC-heS* nanocomposite of *Pnma* Bi_2SeS_2 - *Pnnm* Bi_2SeS_2 are much higher than previously reported Bi_2SeS_2 and Bi_2S_3 -based materials, and other reported sulfide compounds (Fig. 7(e-f)).^{8-11,24,37,42-48} Our work shows that the *hoC-heS* nanocomposite of *Pnma* Bi_2SeS_2 - *Pnnm* Bi_2SeS_2 is a promising TE material. The *hoC-heS* nanocomposite through a dopant-induced phase transition is an effective strategy to decouple the transport of electrons and phonons and boost the TE figure-of-merit."

Fig. R6 Comparison of ZT_{\max} value at the optimum temperature (T_{\max}) of the present $\text{Bi}_2\text{Se}_{1-x}\text{Br}_x\text{S}_2$ ($x=0.12$) with other reported n-type sulfide compounds in the last two

years.¹⁻¹²

References

1. Li, F., et al. *Nano Energy* **88**, 106273 (2021).
2. Bourgès, C., et al. *J. Mater. Chem. C* **8** (46), 16368-16383 (2020).
3. Labégorre, J.-B., et al. *Adv. Funct. Mater.* **29** (48), 1904112 (2019).
4. Guélou, G., et al. *J. Mater. Chem. C* **8** (5), 1811-1818 (2020).
5. Deng, T., et al. *RSC Adv.* **9** (14), 7826-7832 (2019).
6. Liu, J., et al. *Appl. Phys. Lett.* **119** (12), 121905 (2021).
7. Ge, B., et al. *J. Alloys Compd* **809**, 151717 (2019).
8. Bourgès, C., et al. *J. Alloys Compd* **781**, 1169-1174 (2019).
9. Rathore, E., et al. *Chem. Materi.* **31** (6), 2106-2113 (2019).
10. Hu, X., et al. *Scripta Mater.* **170**, 99-105 (2019).
11. Yang, J., et al. *J. Alloys Compd* **780**, 35-40 (2019).
12. Shen, X., et al. *Adv. Funct. Mater.* **30** (21), 2000526 (2020).

Reviewer #3 (Remarks to the Author): *The manuscript reports on the impact of Br doping in Bi₂SeS₂, which induces a structural phase transition from Pnma to Pnmm space group for $x > 0.015$. The carrier concentration is increased by the Br substitution, and moreover the presence of two different nanocomposites is reported to reduce the lattice thermal conductivity. Both effects lead to an enhancement of the ZT values. Compared to previous works, the originality comes from the coexistence of two different crystallographic structures in the same material with almost the same chemical composition. Several state-of-the-art experimental techniques have been used for this work. Moreover, electronic structure calculations and calculations of defects formation energy have been performed to support the hypothesis of Se by Br substitution, and of the role of Br on the electronic gaps. This work is thus interesting and presents a new and elegant approach to optimize ZT values in chalcogenides. There are several points which need to be clarified in the manuscript, points which would deserve a more detailed discussion.*

1. Line 118: is the decrease of lattice parameters really significant? What would be expected from a Vegard's law?

Reply: Thank you. Actually, the decrease of lattice parameters is not significant for the samples with low doped content x ($x < 0.105$), mainly due to the lower dopant

content and small difference in the ionic radius between Br^- (1.96Å) and Se^{2-} (1.98Å). However, when the doped content x higher than 0.105, the lattice parameters gradually reduced, which is consistent with the XRD peaks that gradually shifted toward the higher 2θ as shown in Fig. R7 as following.

Fig. R7 Some main characteristic peaks of the XRD patterns for all the samples

However, the decrease of the lattice parameters seems not strictly follow Vegard's law. One possible reason could be the XRD refinement accuracy limit for the lattice parameter due to the co-existence two phases. Another possible reason should be the existence of microstrain in the lattice owing to the structural distortion in our composite material. In fact, several cases of both positive and negative deviations from the Vegard's law have also been documented in the references (*Adv. Energy Mater.*7(19), 1700573(2017); *Solid State Commun.*322, 114060(2020); *Joule* 2(5), 976-987(2018)). It has shown a large variation and dependence on many physical features, such as the type of conductivity of the sample, depending on the state of the structure: nanostructuring of the sample, a specific volume of a mixed lattice etc. Therefore, the experimental data for the dependence of the lattice parameter differ greatly depending on the structural variations.

In order to make it much clear to the reader, we have added more discussion in the revised manuscript. Details are as follows.

Original: "For the $Pnmm$ phase, the b - and c -axes reduce with increasing Br

content, suggesting that Br enters the lattice (Fig. S4)."

Amended: "For the $Pn\bar{n}m$ phase, the lattice parameters of b and c gradually reduce when the Br content x is higher than 0.105 (Fig. S5). However, the decrease is not significant for the samples with low Br content x ($x < 0.105$), mainly due to the lower dopant content and the slight difference in the ionic radius between Br^- (1.96 Å) and Se^{2-} (1.98 Å)."

2. *The description of Formation Energy in the Supplementary part should be described in more details to justify the use of the chemical formula. And is this formulation coming only from the formation energy calculations or is there some experimental proof of the Br location?*

Reply: Thank you. It is based on the formation energy calculation. As shown in Fig. S11(a) in the revised Supplementary material, the formation energy value for Br doping at Se site is much lower than that at S site. This means Br prefers to occupy the site of Se when doping in Bi_2SeS_2 . Thus, the chemical formula of Br doped samples can use as $\text{Bi}_2\text{Se}_{1-x}\text{Br}_x\text{S}_2$.

Additionally, we also have tried to find the location of Br in the lattice by high-resolution transmission electron microscopy (HRTEM). But it seems so difficult to confirm the Br atoms in the image. We will try our best to observe it in our future study using more advanced transmission electron microscopy.

We have added more descriptions for the formation energy in the revised manuscript. Details are as follows.

Added: " Besides, the calculated formation energy also shows that the formation energy value for Br doping at Se site is much lower than that at S site (Fig. S11(a)). This means Br prefers to occupy the Se site in Bi_2SeS_2 ."

In addition, more descriptions about the calculation of the Formation Energy in detail were also added in the revised Supplementary material. Detail are as follows.

Added: "The formation energy of Bi_2SeS_2 compound, $E_{for}^{\text{Bi}_2\text{SeS}_2}$, is calculated according to Equation S5^{4,5}:"

$$E_{for}^{Bi_2SeS_2} = E_{tot}(Bi_2SeS_2) - 2\mu_{Bi} - \mu_{Se} - 2\mu_S \quad (S5)$$

where $E_{tot}(Bi_2SeS_2)$ is the total energy of the Bi_2SeS_2 compound, and μ_{Bi} , μ_{Se} and μ_S are the chemical potentials of Bi, Se and S, respectively. The chemical potentials are equal to the DFT total energies of their ground states. The formation energy of a Br substituting a Se $\Delta E_{for}(Br_{Se})$ (and S $\Delta E_{for}(Br_S)$), is given by:

$$\Delta E_{for}(Br_{Se}) = E_{tot}(Br_{Se}) - E_{tot}(Bi_2SeS_2) + \mu_{Se} - \mu_{Br} \quad (S6)$$

Where $E_{tot}(Br_{Se})$, $E_{tot}(Bi_2SeS_2)$ are the DFT total energies of the doped and pristine cell, respectively, and μ_{Se} and μ_{Br} are the chemical potential of Se and Br."

3. *Graph 5a: why are the x values for x = 0.12 and x = 0.21 shifted on the x values ?*

Reply: Thank you. We are sorry for this typo. We have checked the original data for the carrier concentration and mobility of all the samples. It is found that the x values for x = 0.12 and x = 0.21 have been mistyped into x=0.13 and x=0.20 when draw the figure. So the x values for x = 0.12 and x = 0.21 shifted on the x values in Graph 5a. We have corrected it in the revised manuscript. The variation trends of the carrier concentration and mobility are not affected.

4. *As shown in S7, the Pisarenko plot fails for x values larger than 0.015. Does it have any impact on the Hall effect curves? Only the n values are given in Fig.5a, but can the authors comment on the extraction of n values by themselves? Is it still a single band model, with a linear behavior of R(H)? These curves should be given in the Supplementary data.*

Reply: Thank you. Fig. R8 as follows shows the variation of the Hall resistance ($R_H=V_H/I$) with magnetic field (B). We can find that R_H shows a linear behavior with the magnetic field for all the samples, indicating the as-fabricated Bi_2SeS_2 nanocomposite can be assumed as a uniform phase using a single parabolic band model for the carrier concentration estimation. Therefore, the Pisarenko relation as exhibited in the Supporting information, which was obtained by assuming single parabolic band model, should be reasonable in the present study.

As predicted by Pisarenko relation from formula S1 to S4 in supporting information, Seebeck coefficient (S) should decrease with increasing n , when m^*_d , λ and T are kept as constants. However, as m^*_d or λ changes (increases), S values will deviate from (enhances as compared with) Pisarenko relation. Therefore, (at a given temperature) whether S value deviates from the Pisarenko relation is a criterion for the change of m^*_d or/and λ . In our *hoC-heS* nanocomposite system, the electronic structure of system is impaired and thus the electronic density of states m^*_d of the system changes with increasing the x content. Thus, the S of Br doped samples with $x \geq 0.015$ above (higher than) the solid line (Pisarenko relation at 300 K) might be ascribed to the increase of effective mass. In fact, we have calculated the effective mass as described in response to Comment 4 of reviewer 2, assuming the current composition is a single uniform phase. Variations in effective mass values (derived from experimental data) at different Br content were observed, which can affect the Seebeck coefficient obviously.

In order to make it much clear, we have added the following Fig. R8 in the revised Supplementary materials as Fig. S10. More discussion about the effective mass was also added in the revised manuscript and supporting information as discibred in the response to the question 4 of the reviewer 2.

Fig. R8 Variation of Hall resistance ($R_H=V_H/I$) with magnetic field (B). (a) Br-free sample Bi_2Se_2 , (b) Br doped samples $\text{Bi}_2\text{Se}_{1-x}\text{Br}_x\text{S}_2$.

5. Line 224: In Figure 7b, it is impossible to read the Callaway fits. The figure caption may be wrong as there are no dotted lines, it rather seems to be straight lines. Also the same symbols should be used for all the curves, and the same as in the supplementary curves (see for example $x=0$ in Fig. 7a and 7b, two different symbols are used).

Reply: Thank you. In Fig. 7(b) in the original manuscript, the dotted line seems to be straight lines because the dots are too dense. We have modified the format for lines of theoretically calculated values of the lattice thermal conductivity (κ_L) to make the readers to read the Callaway fits easily.

Besides, we also have checked and corrected the symbols used for all the curves in the manuscript and the supporting information.

6. Lines 234: the authors should explain why the parameters B and C in the Callaway fits are chosen to remain constant, and how these B and C values were chosen?

Reply: Thank you. To simplify the calculation process, four dominating scattering mechanisms are considered in the present work, which include the impurity/point defect phonon scattering, phonon-phonon Umklapp scattering, electron-phonon scattering and phonon scattering from phase boundaries between *Pnma* Bi₂SeS₂ and *Pnnm* Bi₂SeS₂ phases inside the system. The parameters A, B and C are preset parameters for the relaxation time related to impurity/point defect phonon scattering (τ_{PD}^{-1}), phonon-phonon Umklapp scattering (τ_U^{-1}) and electron-phonon scattering (τ_{EP}^{-1}) inside the system as following.

$$\tau_{PD}^{-1} = \frac{V\Gamma}{4\pi v^3} \omega^4 \propto A\omega^4 \quad (1)$$

$$\tau_U^{-1} = \frac{\hbar\gamma^2\omega^2T}{Mv^2\theta_D} \exp\left(-\frac{\theta_D}{3T}\right) \propto B\omega^2 \quad (2)$$

$$\tau_{EP}^{-1} = \beta\tau_U^{-1} = \beta \frac{\hbar\gamma^2\omega^2T}{Mv^2\theta_D} \exp\left(-\frac{\theta_D}{3T}\right) \propto C\omega^2 \quad (3)$$

Then, the total phonon relaxation time (τ_T) can be simplified as,

$$\tau_T^{-1} = A\omega^4 + B\omega^2T \exp\left(-\frac{\theta_D}{3T}\right) + C\omega^2 + v/L \quad (4)$$

Here, v/L represents phonon scattering from phase boundaries. Hence, for undoped sample, one can obtain parameters A, B and C by substituting formula (4) (but excluding term v/L for the undoped Bi_2SeS_2) for τ_T in Debye model (formula (S8) in the revised supporting information) by fitting experimental data of κ_L for Bi_2SeS_2 to formula (S8).

Then, by using the obtained parameters A, B and C, one can calculate κ_L for the typical Br doped samples with $x \geq 0.015$ through formula (S8) and formula (S12) in the revised supporting information. The calculated results are given (the dotted line) in Fig. 7(b). In this calculation process, phonon-phonon Umklapp scattering (τ_U^{-1}) and electron-phonon scattering (τ_{EP}^{-1}) are kept as constant which are same as those for the undoped sample, because they are independent on induced defects and phase boundaries in present doped samples. The phonon scattering from impurity/point defect (τ_{PD}^{-1}) and phase boundaries (v/L) were mainly taken into account to find the effect of Br doping on lattice thermal conductivity.

In order to make it much clear to the readers, we have added these details in the revised supporting information.

7. *Minor typos: Line 77: 'very different from those', Line 95 : 'predicted structure', Line 105 : 'planar transitions', Line 169 : 'the magnitude of S' rather than 'S', Supplementary : 'energy dependent constant)', Line 253 : 'different from'*

Reply: Thanks a lot. We have corrected all of them. In addition, we have checked the whole paper thoroughly and corrected all the minor errors.

8. *To conclude, the effect of 'homo-composition and hetero-structure' seems to be new and interesting to improve the ZT values of these chalcogenides. The paper is well organized but more details should be given on the description of transport data, to give a more precise description of the fitting parameters and on the impact of 'hetero-structuring' on the electronic transport.*

Reply: Thank you. In our revised manuscript and supporting information, we

have added more details about the description of transport data as mentioned above. In addition, we have also taken on board all the comments given by reviewer 1 and reviewer 2, and calculated the density of state effective mass, the electrical thermal conductivity, and bipolar thermal conductivity, etc. to make the discussion much more clear.

REVIEWERS' COMMENTS

Reviewer #1 (Remarks to the Author):

The authors have responded exhaustively and adequately to the criticisms raised by reviewers and improved the manuscript accordingly. I think the article is clearer now, and I recommend the revised manuscript for publication.

Reviewer #2 (Remarks to the Author):

The authors have revised the whole manuscript according to the reviewers' suggestions. I think this manuscript can be accepted in current version.

Reviewer #3 (Remarks to the Author):

Report on the revised version NCOMMS-21-33758A

The authors have clearly answered the different questions.

There are still some minor points or typos to correct before publication:

Line 64 : the authors should add some references for the donor property of Br.

Line 189 : 'with Br content results from the band edge.'

Line 218 : 'Br should prefer to occupy the Se site'.

Dear Reviewers,

We would like to thank you very much for your comments and suggestions which were extremely helpful to further improve our manuscript (NCOMMS-21-33758A). We have taken on board all these comments and revised the manuscript very closely to what was advised. Please find below our detailed reply to each comment as well as our revised manuscript, where the modified parts are marked blue. We hope the revisions are acceptable.

Thank you very much in advance.

Revised details:

Reviewer #1 (Remarks to the Author): *The authors have responded exhaustively and adequately to the criticisms raised by reviewers and improved the manuscript accordingly. I think the article is clearer now, and I recommend the revised manuscript for publication.*

Reply: Thanks for the referee's time and care in reviewing our manuscript; the comments encourage us a lot.

Reviewer #2 (Remarks to the Author): *The authors have revised the whole manuscript according to the reviewers' suggestions. I think this manuscript can be accepted in current version.*

Reply: Thanks for the referee's time and care in reviewing our manuscript; the comments encourage us a lot.

Reviewer #3 (Remarks to the Author): *The authors have clearly answered the different questions. There are still some minor points or typos to correct before publication:*

Line 64 : the authors should add some references for the donor property of Br.

Line 189 : 'with Br content results from the band edge..'

Line 218 : 'Br should prefer to occupy the Se site'

Reply: Thanks for the referee's time and vigor in reviewing our manuscript. We have corrected all the them. Details are as follows:

In line 189: "with Br content is resulted from the band structure change and the interface effect." has changed into "with Br content results from the band structure change and the interface effect".

In line 218: "Br prefers to occupy the Se site" has changed into "Br should prefer to occupy the Se site".

In addition, we have added two references as follows for the donor property of Br in line 64 in the manucript.

"Guo, J. et al. High thermoelectric properties realized in earth-abundant Bi_2S_3 bulk via carrier modulation and multi-nano-precipitates synergy. *Nano Energy* **78**, 105227 (2020)."

"Liu, Z. H. et al. Enhanced thermoelectric performance of Bi_2S_3 by synergistical action of bromine substitution and copper nanoparticles. *Nano Energy* **13**, 554-562 (2015)."